



# Characterization of Errors in Satellite-based HCHO/NO₂ Tropospheric Column Ratios with Respect to Chemistry, Column to PBL Translation, Spatial Representation, and Retrieval Uncertainties

Amir H. Souri[1]*, Matthew S. Johnson[2], Glenn M. Wolfe[3], James H. Crawford[4], Alan Fried[5], Armin Wisthaler[6,7], William H. Brune[8], Donald R. Blake[9], Andrew J. Weinheimer[10], Tijl Verhoelst[11], Steven Compernolle[11], Gaia Pinardi[11], Corinne Vigouroux[11], Bavo Langerock[11], Sungyeon Choi[3,12], Lok Lamsal[3,13], Lei Zhu[14,15], Shuai Sun[14,15], Ronald C. Cohen[16,17], Kyung-Eun Min[18], Changmin Cho[18], Sajeev Philip[19], Xiong Liu[1], and Kelly Chance[1]

[1]Atomic and Molecular Physics (AMP) Division, Center for Astrophysics | Harvard & Smithsonian, Cambridge, MA, USA
[2]Earth Science Division, NASA Ames Research Center, Moffett Field, CA, USA
[3]NASA Goddard Space Flight Center, Greenbelt, MD, USA
[4]NASA Langley Research Center, Hampton, VA, USA
[5]Institute of Arctic & Alpine Research, University of Colorado, Boulder, CO, USA
[6]Institute for Ion Physics and Applied Physics, University of Innsbruck, Technikerstrasse 25, 6020 Innsbruck, Austria
[7]Department of Chemistry, University of Oslo, P.O. box 1033, Blindern, 0315 Oslo, Norway
[8]Department of Meteorology and Atmospheric Science, Pennsylvania State University, University Park, PA, USA
[9]Department of Chemistry, University of California, Irvine, CA, USA
[10]National Center for Atmospheric Research, Boulder, CO, USA
[11]Royal Belgian Institute for Space Aeronomy (BIRA-IASB), Ringlaan 3, 1180 Uccle, Belgium
[12]Science Systems and Applications, Inc., Lanham, MD 20706, USA
[13]Universities Space Research Association, Columbia, MD 21046, USA
[14]School of Environmental Science and Engineering, Southern University of Science and Technology, Shenzhen, Guangdong, China
[15]Guangdong Provincial Observation and Research Station for Coastal Atmosphere and Climate of the Greater Bay Area, Shenzhen, Guangdong, China
[16]Department of Earth and Planetary Science, University of California Berkeley, Berkeley, CA 94720, USA
[17]Department of Chemistry, University of California Berkeley, Berkeley, CA 94720, USA
[18]School of Earth Sciences and Environmental Engineering, Gwangju Institute of Science and Technology, Gwangju, South Korea
[19]Centre for Atmospheric Sciences, Indian Institute of Technology Delhi, New Delhi, India

* Corresponding Author: ahsouri@cfa.harvard.edu

**Abstract.**

The availability of formaldehyde (HCHO) (a proxy for volatile organic compound reactivity) and nitrogen dioxide (NO₂) (a proxy for nitrogen oxides) tropospheric columns from Ultraviolet-Visible (UV-Vis) satellites has motivated many to use their ratios to gain some insights into the near-surface ozone sensitivity. Strong emphasis has been placed on the challenges that come with transforming what is being observed in the tropospheric column to what is actually in the planetary boundary layer (PBL) and near to the surface; however, little attention has been paid


to other sources of error such as chemistry, spatial representation, and retrieval uncertainties. Here
we leverage a wide spectrum of tools and data to carefully quantify those errors.
Concerning the chemistry error, a well-characterized box model constrained by more than
500 hours of aircraft data from NASA's air quality campaigns is used to simulate the ratio of the
chemical loss of $HO_2+RO_2$ (LROx) to the chemical loss of $NO_x$ (LNOx). Subsequently, we
challenge the predictive power of $HCHO/NO_2$ ratios (FNRs), which are commonly applied in
current research, at detecting the underlying ozone regimes by comparing them to LROx/LNOx.
FNRs show a strongly linear ($R^2=0.94$) relationship to LROx/LNOx in the log-log scale. Following
the baseline (i.e., ln(LROx/LNOx) = -1.0±0.2) with the model and mechanism (CB06, r2) used for
segregating NOx-sensitive from VOC-sensitive regimes, we observe a broad range of FNR
thresholds ranging from 1 to 4. The transitioning ratios strictly follow a Gaussian distribution with
a mean and standard deviation of 1.8 and 0.4, respectively. This implies that FNR has an inherent
20% standard error (1-sigma) resulting from not being able to fully describe the ROx-HOx cycle.
We calculate high ozone production rates (PO₃) dominated by large $HCHO \times NO_2$ concentration
levels, a new proxy for the abundance of ozone precursors. The relationship between PO₃ and
$HCHO \times NO_2$ becomes more pronounced when moving towards NOx-sensitive regions due to non-
linear chemistry; our results indicate that there is fruitful information in the $HCHO \times NO_2$ metric
that has not been utilized in ozone studies. The vast amount of vertical information on HCHO and
NO₂ concentration from the air quality campaigns enables us to parameterize the vertical shapes
of FNRs using a second-order rational function permitting an analytical solution for an altitude
adjustment factor to partition the tropospheric columns to the PBL region. We propose a
mathematical solution to the spatial representation error based on modeling isotropic
semivariograms. With respect to a high-resolution sensor like TROPOspheric Monitoring
Instrument (TROPOMI) (>5.5×3.5 km²), Ozone Monitoring Instrument (OMI) loses 12% of
spatial information at its native resolution. A pixel with a grid size of 216 km² fails at capturing
~65% of the spatial information in FNRs at a 50 km length scale comparable to the size of a large
urban center (e.g., Los Angeles). We ultimately leverage a large suite of in-situ and ground-based
remote sensing measurements to draw the error distributions of daily TROPOMI and OMI
tropospheric NO₂ and HCHO columns. At 68% confidence interval (1 sigma) errors pertaining to
daily TROPOMI observations, either HCHO or tropospheric NO₂ columns should be above 1.2-
$1.5 \times 10^{16}$ molec.cm$^{-2}$ to attain 20-30% standard error in the ratio. This level of error is almost non-
achievable with OMI given its large error in HCHO.
The satellite column retrieval error is the largest contributor to the total error (40-90%) in
the FNRs. Due to a stronger signal in cities, the total relative error (<50%) tends to be mild,
whereas areas with low vegetation and anthropogenic sources (e.g., Rocky Mountains) are
markedly uncertain (>100%). Our study suggests that continuing development in the retrieval
algorithm and sensor design and calibration is essential to be able to advance the application of
FNRs beyond a qualitative metric.

## 1. Introduction

Accurately representing the near-surface ozone (O₃) sensitivity to its two major precursors,
nitrogen dioxides (NOx) and volatile organic compounds (VOCs), is an imperative step in
understanding non-linear chemistry associated with ozone production rates in the atmosphere.
While it is often tempting to characterize an airshed as NOx or VOC-sensitive, both conditions are
expected as VOC-sensitive conditions near NOx sources transition to NOx-sensitive conditions
downwind as NOx dilutes. Thus, reducing the footprint of ozone production can mostly be
achieved through NOx reductions. VOCs are key to determining both the location and peak in



ozone production which varies nonlinearly to the NOx abundance. Thus, knowledge of the relative
levels of NOx and VOCs informs the trajectory of ozone production and expectations of where
peak ozone will occur as emissions change. While the near-surface ambient nitrogen dioxide ($NO_2$)
concentrations are regularly monitored by a large number of surface stations, the measurements of
several VOCs with different reactivity rates with respect to hydroxyl (OH), are not routinely
available. As such, our knowledge on where and when ozone production rates are elevated, and
their quantitative dependence on a long list of ozone precursors, is fairly limited, except for
observationally-rich air quality campaigns. This limitation has prompted several studies such as
Sillman et al. (1990), Tonnesen and Dennis (2000a,b), and Sillman and He (2002) to investigate if
the ratio of certain measurable compounds can diagnose ozone regimes meaning if the ozone
production rate is sensitive to NOx (i.e., NOx-sensitive) or VOC (i.e., VOC-sensitive). Sillman
and He (2002) suggested that $H_2O_2/HNO_3$ was a robust measurable ozone indicator as this ratio
could well describe the chemical loss of $HO_2+RO_2$ (LROx) to the chemical loss of NOx (LNOx)
controlling the $O_3$-NOx-VOC chemistry (Kleinman et al., 2001). Nonetheless, both $H_2O_2$ and
$HNO_3$ measurements are limited to few spatially-sparse air quality campaigns.
Formaldehyde (HCHO) is an oxidation product of VOCs and its relatively short lifetime
(~1-9 hr) makes the location of its primary and secondary sources rather identifiable (Seinfield and
Pandis, 2006; Fried et al., 2020). Fortunately, monitoring HCHO abundance in the atmosphere has
been a key goal of many Ultraviolet-Visible (UV-Vis) viewing satellites for decades (Chance et
al., 1991; Chance et al., 1997; Chance et al., 2000; González Abad et al., 2015; De Smedt et al.,
2008, 2012, 2015, 2018, 2021) with reasonable spatial coverage. Additionally, the strong
absorption of $NO_2$ in the UV-Vis range has permit measurements of $NO_2$ columns from space
(Martin et al., 2002; Boersma et al., 2004, 2007, 2018).
Advancements in satellite remote-sensing of these two key compounds have encouraged
many studies to elucidate if the ratio of $HCHO/NO_2$ (hereafter FNR) could be a robust ozone
indicator (Tonnensen and Dennis, 2000b; Martin et al. 2004, Duncan et al., 2010). Most studies
using the satellite-based FNR columns attempted to provide a qualitative view of the underlying
chemical regimes (e.g., Choi et al., 2012; Choi and Souri, 2015a,b; Jin and Holloway, 2015; Souri
et al., 2017; Jeon et al., 2018; Lee et al., 2021). Relatively few studies (Duncan et al., 2010; Jin et
al., 2017; Schroeder et al., 2017; Souri et al., 2020) have carefully tried to provide a quantitative
view of the usefulness of the ratio. For the most part, the inhomogeneous vertical distribution of
FNR in columns has been emphasized. Jin et al. (2017) and Schroeder et al. (2017) showed that
differing vertical shapes of HCHO and $NO_2$ can cause the vertical shape of FNR not to be
consistent throughout the troposphere leading to a variable relationship between what is being
observed from the satellite and what is actually occurring in the lower atmosphere. Jin et al. (2017)
calculated an adjustment factor to translate the column to the surface using a relatively coarse
global chemical transport model. The adjustment factor showed a clear seasonal cycle stemming
from spatial and temporal variability associated with the vertical sources and sinks of HCHO and
$NO_2$, in addition to the atmospheric dynamics. In a more data driven approach, Schroeder et al.
(2017) found that the detailed differences in the boundary layer vertical distributions of HCHO
and $NO_2$ lead to a wide range of ambiguous ratios. Additionally, ratios were shown to shift on high
ozone days, raising questions regarding the value of satellite averages over longer timescales. A
goal for our research is to put together an integrated and data-driven mathematical formula to
translate the tropospheric column to the PBL, exploiting the abundant aircraft measurements
available during ozone seasons.



Using observationally-constrained box models, Souri et al. (2020) demonstrated that there
was a fundamentally inherent uncertainty related to the ratio originating from the chemical
dependency of HCHO on $NO_x$ (Wolfe et al., 2016). In VOC-rich (poor) environments, the
transitioning ratios from NOx-sensitive to VOC-sensitive occurred in larger (smaller) values than
the conventional thresholds defined in Duncan et al. (2010) due to an increased (dampened) HCHO
production induced by NOx. To account for the chemical feedback and to prevent a wide range of
thresholds on segregating NOx-sensitive from VOC-sensitive regions, Souri et al. (2020)
suggested using a first-order polynomial matched to the ridgeline in $P(O_3)$ isopleths. Their study
illuminated the fact that the ratio suffers from an inherit chemical complication. However, Souri
et al. (2020) did not quantify the error and their work was limited to a subset of atmospheric
condition. To challenge the predictive power of FNR from chemistry perspective, we will take
advantage of a large suite of datasets to make maximum use of varying meteorological and
chemical conditions.
Not only are satellite-based column measurements unable to resolve the vertical
information of chemical species in the tropospheric column, but they are also unable to resolve the
horizontal spatial variability due to their spatial footprint. The larger the footprint is, the more
horizontal information is blurred out. For instance, Souri et al. (2020) observed a substantial spatial
variance (information) in FNR columns at the spatial resolution of $250 \times 250$ m$^2$ observed by an
airborne sensor over Seoul, South Korea. It is intuitively clear that a coarse resolution sensor would
lose a large degree of spatial variance (information). This error, known as the spatial representation
error, has not been studied with respect to FNR. We will leverage what we have learned from Souri
et al. (2022), which modeled the spatial heterogeneity in discrete data using geostatistics, to
quantify the spatial representation error in the ratio over an urban environment.
A longstanding challenge is to have a reliable estimate on the satellite retrieval errors of
tropospheric column $NO_2$ and HCHO. Significant efforts have been made recently to assemble,
analyze, and estimate the retrieval errors for two key satellite sensors, TROPOspheric Monitoring
Instrument (TROPOMI) and Ozone Monitoring Instrument (OMI), using various in-situ
measurements (Verhoelst et al., 2021; Vigouroux et al., 2020, Choi et al., 2020; Laughner et al.,
2019; Zhu et al., 2020). In this study, we will exploit paired comparisons from some of these new
studies to propagate individual uncertainties in HCHO and $NO_2$ to the FNR errors.
The overarching science goal of this study is to address the fact that the accurate diagnosis
of surface $O_3$ photochemical regimes is impeded by numerous uncertainty components, which will
be addressed in the current paper, and can be classified into four major categories: i) inherent
uncertainties associated with the approach of FNRs to diagnose local $O_3$ production and sensitivity
regimes, ii) translation of tropospheric column satellite retrievals to represent PBL- or surface-
level chemistry, iii) spatial representativity of ground pixels of satellite sensors, and iv)
uncertainties associated with satellite-retrieved column-integrated concentrations of HCHO and
$NO_2$. We will address all of these sources of uncertainty using a broad spectrum of data and tools.
Our paper is organized with the following sections. Section 2 describes the chemical box
model setup and data applied. Sections 3.1 to 3.4 deal with chemistry aspects of FNRs and show
the results from the box model. Section 3.5 introduces a data-driven framework to transform the
FNR tropospheric columns to the PBL region. Section 3.6 offers a new way to quantify the spatial
representation error in satellites. Section 3.7 deals with the satellite error characterization and their
impacts on the ratio. Section 3.8 summarizes the fractional contribution of each error to the total
error. Finally, Sect. 4 provides a summary and conclusions of the study.



## 2. Photochemical Box Modeling and Aircraft Data Used

To quantify the uncertainty of FNR from a chemical perspective, and to obtain several imperative parameters including the calculated ozone production rates, and the loss of $NO_x$ ($LNO_x$) and $RO_x$ ($LRO_x$), we utilize the Framework for 0-D Atmospheric Modeling (F0AM) v4 (Wolfe et al., 2016). We adopt the Carbon Bond 6 (CB06, r2) chemical mechanism and heterogenous chemistry is not considered in our simulations. The model is initialized with the measurements of several compounds, many of which constrain the model by being held constant for each timestep (see Table 1). It is important to acknowledge that the VOC constraints for these model calculations are incomplete, especially for the DISCOVER-AQ campaigns which lacked comprehensive VOC observations. Nevertheless, we will show that the selected VOCs are sufficient to reproduce a large variance (>70%) in observed HCHO.

Figure 1 shows the map of data points from Deriving Information on Surface Conditions from Column and Vertically Resolved Observations Relevant to Air Quality (DISCOVER-AQ) Baltimore-Washington (2011), DISCOVER-AQ Houston-Texas (2013), DISCOVER-AQ Colorado (2014), and Korea United States Air Quality Study (KORUS-AQ) (2016). Meteorological inputs come from the observed pressure, temperature, and relative humidity. The measurements of photolysis rates are not available for all photolysis reactions; therefore, our initial guess of those rates comes from a look-up-table populated by the National Center for Atmospheric Research (NCAR) Tropospheric Ultraviolet And Visible (TUV) model calculations. These values are a function of solar zenith angle, total ozone column density, surface albedo, and altitude. We set the total ozone column and the surface albedo to fixed numbers of 325 (Dobson) DU and 0.15, respectively. The initial guess is then corrected by applying the ratio of observed photolysis rates of $NO_2$+hv ($jNO_2$) and/or $O_3$+hv ($jO^1D$) to the calculated ones to all j-values (i.e., wavelength independent). If both observations of $jNO_2$ and $jO^1D$ are available, the correction factor is averaged. The KORUS-AQ campaign is the only one that provides $jO^1D$ measurements; therefore the use of the wavelength-independent correction factor based on the observed to calculated $jNO_2$ values for all j-values including $jO^1D$ is a potential source of error in the model especially when aerosols are present. The model calculations are based on the observations merged to a temporal resolution varying from 10 to 15 seconds. Each calculation was run for five consecutive days with an integration time of 1 hour to approach diel steady state. Some secondarily-formed species must be unconstrained for the purpose of model validation. Therefore, the concentrations of several secondarily-formed compounds such as HCHO and PAN are unconstrained. Nitric oxide (NO) and $NO_2$ are also allowed to cycle, while their sum (i.e., $NO_x$) is constrained. Because the model does not consider various physical loss pathways including deposition and transport, we oversimplify their physical loss through a first-order dilution rate set to $1/86400$-$1/43200$ s$^{-1}$ (i.e., 24- or 12-hr lifetime), which in turn prevents relatively long-lived species from accumulating over time. The optimal dilution rate is determined to ensure a marginal difference between the average of simulated HCHO and observations (<5%) for each air quality campaign. The dilution factor is set to a fixed value for an entire campaign. Each time tag is independently simulated meaning we do not initialize the next run using the simulated values from the previous one; this in turn permits parallel computation. Table 1 lists the major configuration along with the observations used for the box model.

Several parameters are calculated based on the box model outputs. $LRO_x$ is defined through the sum of primarily radical-radical reactions:





$$LRO_x = k_{HO_2+HO_2}[HO_2]^2 + \sum k_{RO_{2i}+HO_2}[RO_{2i}][HO_2] + \sum k_{RO_{2i}+RO_{2i}}[RO_{2i}]^2 \tag{1}$$

$LNO_x$ mainly occurs via the $NO_2+OH$ reaction:

$$LNO_x = k_{OH+NO_2+M}[OH][NO_2][M] \tag{2}$$

We calculate $P(O_3)$ by subtracting the ozone loss pathways dictated by $HO_x$ ($HO+HO_2$), $NO_2+OH$,
$O_3$ photolysis, ozonolysis, and the reaction of $O(^1D)$ with water vapor from the formation pathways
through the removal of NO via $HO_2$ and $RO_2$:

$$P(O_3) = k_{HO_2+NO}[HO_2][NO] + \sum k_{RO_{2i}+NO}[RO_{2i}][NO] \\ - k_{OH+NO_2+M}[OH][NO_2][M] - P(RONO_2) - k_{HO_2+O_3}[HO_2][O_3] \\ - k_{OH+O_3}[OH][O_3] - k_{O(^1D)+H_2O}[O(^1D)][H_2O] - L(O_3 \\ + alkenes) \tag{3}$$

### 232  3. Results and Discussion

### 233  *3.1.  Box Model Validation*

There are uncertainties associated with the box model (e.g., Brune et al., 2021; Zhang et
al., 2021; Lee et al., 2021) which can be attributed to: i) the lack of inclusion of physical processes
such as entrainment/detrainment, ii) discounting the heterogenous chemistry, iii) invalid
assumption of the diel steady state in areas close to large emission sources or in photochemically
less active environments (Thornton et al., 2002; Souri et al., 2021), iv) errors in the chemical
mechanism, and v) errors in the measurements. These limitations necessitate a thorough validation
of the model using unconstrained observations. While models have been known for a long time to
not be 100% accurate (Box, 1976), it is important to characterize whether the model can effectively
represent reality. For instance, if the simulated HCHO is poorly correlated with observations
and/or displayed large magnitude biases, it will be erroneous to assume that the sources of HCHO
along with relevant chemical pathways are appropriate.
We diagnose the performance of the box model by comparing the simulated values of five
compounds to observations: HCHO, NO, $NO_2$, PAN, hydroperoxyl radical ($HO_2$), and OH. Figure
2 depicts the scatterplot of the comparisons along with several statistics. HCHO observations are
usually constrained in box models to improve the representation of $HO_2$ (Schroeder et al., 2017;
Souri et al., 2020; Brune et al., 2021); however, this constraint may mask the realistic
characterization of the chemical mechanism with respect to the treatment of VOCs. Additionally,
it is important to know if the sources of HCHO are adequate. Therefore, we detach the model from
this constraint to carry out a more fair and stringent validation. Concerning HCHO, our model
does have considerable skill at reproducing the variability of observed HCHO ($R^2=0.73$) with a
low bias of -4.9% (-0.09 ppbv). Likewise, the model performs well with regards to the simulation
of NO ($R^2=0.89$) and $NO_2$ ($R^2=0.99$) in the log scale. Immediately evident is the underestimation
of NO in highly polluted regions contrary to overestimation in clean ones. This discrepancy leads
to an underestimation (overestimation) of $NO/NO_2$ in polluted (clean) regions. The primary drivers
of $NO/NO_2$ are $jNO_2$ and $O_3$ both of which are constrained in the model. What can essentially
deviate the partitioning between NO and $NO_2$ from that of observations in polluted areas is the
assumption of the diel steady state which is rarely strictly valid where measurements are close to
large emitters. The overestimation of NO in low $NO_x$ areas is often blamed on the lack of chemical





sink pathways of NO in chemical mechanism (e.g., Newland et al., 2021). A relatively reasonable
performance of PAN ($R^2$=0.63) is possibly due to constraining some of oxygenated VOCs such as
acetaldehyde. Xu et al. (2021) observed a strong dependency of PAN concentrations on $NO/NO_2$
ratios. Smaller $NO/NO_2$ ratios are usually associated with larger PAN because NO can effectively
remove peroxyacetyl radicals. We observe an overestimation of PAN (0.27 ppbv) possibly due to
an underestimation of $NO/NO_2$. Moreover, we should not rule out the impact of the first-order
dilution factor which was only empirically set in this study. For instance, if we ignore the dilution
process, the bias of the model in terms of PAN will increase by 33% resulting in a poor
performance ($R^2$=0.40) (not shown). Schroeder et al. (2020) found that proper simulation of PAN
in the polluted PBL during KORUS-AQ required a first-order loss rate based on thermal
decomposition at the average PBL temperature, which was more realistic than the widely varying
local PAN lifetimes associated with temperature gradients between the surface and the top of the
PBL. This solution is computationally equivalent to the dilution rate used in this study.
KORUS-AQ was the only field campaign providing OH and $HO_2$ measurements.
Concerning $HO_2$, former studies such as Schroeder et al. (2017), Souri et al. (2020), and Brune et
al. (2021) managed to reproduce $HO_2$ with $R^2$ ranging from 0.6 to 0.7. The performance of our
model ($R^2$=0.66) is similar to these past studies with near negligible biases (<1%). One may argue
that the absence of the $HO_2$ uptake by aerosols is contributing to some of the discrepancies we
observe in the $HO_2$ comparison. Brune et al. (2021) provided compelling evidence showing that
the consideration of the $HO_2$ uptake made their results significantly inconsistent with the
observations suggesting that the $HO_2$ uptake might have been inconsequential during the
campaign. Our model manages to reproduce 64% of the variance of observed OH outperforming
the simulations presented in Souri et al. (2020) and Brune et al. (2021) by >10%. The slope (=
1.03) is not too far from the identity line indicating that our box model systematically overestimates
OH by $0.62\ 10^6\ cm^{-3}$. This may be attributed to a missing OH sink in the mechanism or the lack
of inclusion of some VOCs. In general, the model performance is consistent, or outperforms,
results from recent box modelling studies which is an indication of it being at least virtually
representative of the real-world ozone chemistry and sensitivity regimes.

### 3.2. *Can HCHO/NO₂ ratios fully describe the HOx-ROx cycle?*

Kleinman et al. (2001) provided an analytical solution suggesting LROx/LNOx is the most
robust ozone regime indicator. Thus, the predictive power of FNR at detecting the underlying
chemical conditions can be challenged by comparing FNR to LROx/LNOx. Ideally, if they show
a strong degree of correspondence (i.e., $R^2$=1.0), we can confidently say that FNR can realistically
portray the chemical regimes. Any divergence of these two quantities is indicative of inadequacy
of the FNR indicator. Souri et al. (2020) observed a strong linear relationship between the
logarithmic transformed FNR and those of LROx/LNOx. Our analysis in this study will be based
upon the simulated values to ensure that the relationship is coherent based on a realization from
the well-characterized box model. As pointed out by Schroeder et al. (2017) and Souri et al. (2020),
a natural logarithm of LROx/LNOx roughly equal to -1.0 (i.e., LROx/LNOx = 0.35-0.37)
perceptibly separates VOC-sensitive from NOx-sensitive regimes, which would make this
threshold the baseline of our analysis.
Figure 3 demonstrates the log-log relationship of LROx/LNOx and FNR, and $P(O_3)$, from
all four air quality campaigns. The log-log relationships from each individual campaign are shown
in Figure S1-S4. We overlay the LROx/LNOx baseline threshold along with two commonly used
thresholds for FNR suggested by Duncan et al. (2010); they defined the VOC-sensitive regimes if
FNR<1 and the $NO_x$-sensitive ones if FNR>2. Any region undergoing a value between these



thresholds is unlabeled and considered to be in a transitional regime. The size of each data point
is proportional to the HCHO×NO$_2$ concentration magnitude. One striking finding from this plot is
that there is indeed a strong linear relationship between the logarithmic-transformed LROx/LNOx
and FNR (R$^2$=0.91). A strong linear relationship between the two quantities in the log-log scale is
indicative of a power law dependence (i.e., $y=ax^b$). A strong power law dependency means that
these two quantities have a poor correlation at their low and high values. This is mainly caused by
the fact that HCHO does not fully describe VOC reactivity rates in rich and poor VOC
environments (Souri et al., 2020). A question is what range of FNR will fall in ln(LROx/LNOx) =
-1.0±0.2? Following the baseline, the transitioning ratios follow a normal distribution with a mean
of 1.8, a standard deviation of 0.4, and a range from 1 to 4 (Figure S5). We define the chemical
error in the application of FNR to separate the chemical regimes as the relative error standard
deviation (i.e., σ/µ) of the transitioning ratios leading to ~ 20%. These numbers are based on a
single model realization and can change if a different mechanism is used; nonetheless, the model
has considerable skill at reproducing many different unconstrained compounds, especially OH,
suggesting that it is a rather reliable realization. The comparison of the transitioning FNRs to the
NO$_2$ concentrations suggests no correlation (r=0.02) whereas there is a linear correlation between
the transitioning ratios and the HCHO concentrations (r=0.56). This tendency reinforces the study
of Souri et al. (2020) who, primarily due to the HCHO-NO$_2$ feedback, observed a larger FNR
threshold in VOC-rich environments to be able to detect the chemical regimes.

### 3.3. Large PO$_3$ rates occur in regions with large HCHO×NO$_2$ concentrations when moving towards NOx-sensitive regions

A striking and perhaps intuitive tendency observed from Figure 3 is that large PO$_3$ rates
are mostly tied to higher HCHO×NO$_2$. But this relationship gradually weakens as we move
towards VOC-sensitive regions (smaller LROx/LNOx ratios). This is a textbook example of non-
linear ozone chemistry. In VOC-sensitive areas, PO$_3$ can be strongly inhibited by NO$_2$+OH and
the formation of organic nitrates despite the abundance of the precursors. In application of remote-
sensing of ozone precursors, the greatest unused metric describing the mass of the ozone precursors
is HCHO×NO$_2$. However, this metric should only be used in conjunction with FNR. To
demonstrate this, based on what the baseline (LROx/LNOx) suggests against thresholds on FNRs
defined by Duncan et. al. (2010), we group the data into four regions namely as NOx-sensitive –
NOx-sensitive, NOx-sensitive–transitional, VOC-sensitive–transitional, and VOC-sensitive–
VOC-sensitive. A different perspective into this categorization is that the transitional regimes are
a weaker characterization of the main regime; for instance, NOx-sensitive–transitional regions are
less NOx-sensitive than NOx-sensitive – NOx-sensitive. Subsequently, the cumulative distribution
functions (CDFs) of PO$_3$ and HCHO×NO$_2$ with respect to the aforementioned groups are
calculated, which is shown in Figure 4. Regarding NOx-sensitive—NOx-sensitive regions, we see
the PO$_3$ CDF very quickly converging to the probability of 100% indicating that the distribution
of PO$_3$ is skewed towards very low values. The median of PO$_3$ for this particular regime (where
CDF = 50%) is only 0.25 ppbv/hr. This agrees with previous studies such as Martin et al. (2002),
Choi et al. (2012), Jin et al. (2017), and Souri et al. (2017) reporting that NOx-sensitive regimes
dominate in pristine areas. The PO$_3$ CDFs between NOx-sensitive—transitional and VOC-
sensitive—VOC-sensitive are not too distinct, whereas their HCHO×NO$_2$ CDFs are substantially
different. The non-linear ozone chemistry suppresses PO$_3$ in highly VOC-sensitive areas such that
those values are not too different than those in mildly polluted areas (NOx-sensitive—transitional).
Perhaps the most interesting conclusion from this figure is that elevated PO$_3$ values (median = 4.6
ppbv/hr), a factor of two larger than two previous regimes, are mostly found in VOC-sensitive—



transitional. This is primary due to two causes: i) this particular regime is not strongly inhibited by
the nonlinear chemistry, particularly $NO_2+OH$, and ii) it is associated with abundant precursors
evident in the median of $HCHO \times NO_2$ being as three times as large of those in NOx-sensitive—
transitional. This tendency illustrates the notion of the non-linear chemistry and how this may
affect regulations. Simply knowing where the regimes are might not suffice to pinpoint the peak
of $PO_3$, as this analysis suggests that we need to take both FNR and $HCHO \times NO_2$ into
consideration; both metrics are readily accessible from satellite remote-sensing sensors.

### 361     3.4.     *Can we estimate $PO_3$ using the information from $HCHO/NO_2$ and $HCHO \times NO_2$?*

It may be advantageous to construct an empirical function fitted to these two quantities and
elucidate the maximum variance (information) we can potentially gain to recreate $PO_3$. After
several attempts, we found a bilinear function ($z=a_0+a_1x+a_2y+a_3xy$) to be a good fit without
overparameterization. Due to presence of extreme values in both FNR and $HCHO \times NO_2$, we use a
weighted least squares method for the curve fitting based on the distance of the fitted curve to the
data points (known as bi-squares weighting). The best fit with $R^2$ equals to 0.94 and an RMSE of
0.60 ppbv/hr is:

$$PO_3 = 0.74 - 0.09\,x - 0.02\,y + 0.25xy \tag{4}$$

where $x$ and $y$ are FNR (unitless) and $HCHO \times NO_2$ ($ppbv^2$), respectively. The residual of the fit is
shown in Figure S6. The gradients of $PO_3$ with respect to $x$ and $y$ are:

$$\frac{dPO_3}{dx} = 0.25y - 0.09 \tag{5}$$

$$\frac{dPO_3}{dy} = 0.25x - 0.02 \tag{6}$$

An apparent observation arises from these equations that is the derivatives of $PO_3$ to each
metric depends on the other one underscoring their interconnectedness. For instance, Eq. (6)
suggests that larger FNRs ($x$) result in a larger gradient of $PO_3$ to the abundance of $HCHO \times NO_2$
($y$). In very low FNRs, this gradient can become very small rendering $PO_3$ insensitive (or in
extreme cases, negatively correlated) to $HCHO \times NO_2$. This analysis provides encouraging results
about the future application of the satellite-derived $HCHO \times NO_2$; however, the wide class of
problems relating to the application of satellite-derived FNR columns such as satellite errors in
columns or the translation between columns to PBL are also present in Eq. (4), even in a more
pronounced way due to $HCHO \times NO_2$ and $HCHO^2$ ($= xy$). This new perspective into $PO_3$ estimation
deserves a separate study.

### 381     3.5.     *Altitude dependency and its parametrization*

A lingering concern over the application of satellite-based FNR tropospheric columns is
that the vertical distribution of HCHO and $NO_2$ are integrated in columns thus this vertical
information is permanently lost. As such, here we provide insights on the vertical distribution of
FNR within the tropospheric column. This task requires information about the differences between
the vertical shape of HCHO and that of $NO_2$. Ideally, if both compounds show an identically
relative shape, the FNR columns will be valid for every air parcel along the vertical path (i.e., a
straight line). We do not always know the precise knowledge of HCHO and $NO_2$ vertical
distributions, but we can constitute some degree of generalizations by leveraging the
measurements made during the aircraft campaigns.
Figure 5 demonstrates the violin plot of the afternoon (> 12:00 LST) vertical distribution
of HCHO, $NO_2$, and FNR observed by NASA's aircrafts during the four field campaigns analyzed
in this study superimposed by the simulated $PO_3$ rates. The vertical layers are grouped into sixteen





altitudes ranging from 0.25 km to 7.75 km. Each vertical layer incorporates measurements ±0.25
km of the altitude mid-layer height. The observations do not follow a normal distribution,
particularly in the lower parts of the atmosphere; thus medians are preferred to represent the central
tendency. While the largest $PO_3$ rates tend to occur in areas close to the surface (< 2 km agl), a
nonnegligible fraction of the elevated $PO_3$ rates are also observed in other parts of the atmosphere
such as those in the free troposphere.
Several intriguing features are observed from Figure 5: First, up to the 5.75 km range,
which encompasses the PBL area and a large portion of the free-troposphere, $NO_2$ concentrations
tend to decrease quicker than those of HCHO in line with previous studies such as Schroeder et al.
(2017), Jin et al. (2017), Chan et al. (2019), and Ren et al. (2022). Second, above 5.75 km, HCHO
continues to decrease whereas $NO_2$ shows an increasing trend. As a result of their different vertical
trends, we observe nonuniformities in the vertical distribution of FNR: they become more NOx-
sensitive with altitude up to a turning point at 5.75 km and then shift backwards to VOC-sensitive.
It is attractive to model these shapes and apply parameterizations to understand how their
shapes will complicate the use of tropospheric column retrieval from satellites. First order rational
functions are a good candidate to use. Concerning the vertical dependency of HCHO and $NO_2$, we
find reasonable fit ($R^2$=0.73) as:

$$HCHO, NO_2 = \frac{a_0 z + a_1}{z + a_2} \tag{7}$$

where $z$ is altitude in km. $a_i$ ($i$=0,1,2) are fitting parameters. From this equation it is determined
that FNRs follow a second order rational function:

$$f(z) = \frac{HCHO}{NO_2} = \frac{b_0 z^2 + b_1 z + b_2}{b_3 z^2 + b_4 z + b_5} \tag{8}$$

where $b_i$ ($i$=0, ... , 5) are fitting parameters. One can effortlessly fit this function to different bounds
of the vertical distribution of FNR such as the 25$^{th}$ and 75$^{th}$ percentiles, and subsequently estimate
the first moment of the resultant polygon along $z$ via:

$$\left. \frac{\overline{HCHO}}{NO_2} \right|_{z1}^{z2} = \frac{1}{2A} \int_{z1}^{z2} f^2(z)_{75th} - f^2(z)_{25th} \tag{9}$$

where $A$ is the area of the polygon bounded by the 75$^{th}$ percentiles, $f(z)_{75th}$, and the 25$^{th}$
percentiles ($f(z)_{25th}$) of FNR (shown in Figure 5 as solid black lines). We define an altitude
adjustment factor ($f_{adj}$) such that one can translate an observed FNR tropospheric column ratios,
such as those retrieved from satellites, to a defined altitude and below that point ($zt$) through:

$$f_{adj} = \frac{\left. \dfrac{\overline{HCHO}}{NO_2} \right|_0^{zt}}{\left. \dfrac{\overline{HCHO}}{NO_2} \right|_0^{8\,km}} \tag{10}$$

where $zt$ can be interchanged to match the PBLH. This definition is more beneficial than using the
entire tropospheric column to the surface conversion (e.g., Jin et al., 2017) because ozone can be
formed in various vertical layers. Using the observations collected during the campaign, we
estimate Eq. (10) along with ±1σ boundaries shown in Figure 6. The shape of the resulting
adjustment factor is in line with of the vertical distribution of FNR (see Figure 5): the adjustment
factor curve closer to the surface have values smaller than one, increases to values larger than one
in the mid-troposphere, and finally converges to one near the top of the tropospheric column. If
one picks out an altitude pertaining to a PBLH, they can easily apply $f_{adj}$ to the observed FNR
columns to estimate the corresponding ratio for that specific PBLH. A more evolved PBLH (i.e.,





a large $zt$) results in stronger vertical mixing rendering $f_{\text{adj}}$ closer to one. The standard error
deviation of this conversion is around 26%.
It is beneficial to model this curve to make this data-driven conversion easier for future
applications. A second-order polynomial can well describe ($R^2$=0.99) this curve:

$$f_{\text{adj}} = az_t^2 + bz_t + c \qquad\qquad a = -0.02, b = 0.25, c = 0.41 \qquad\qquad (11)$$

Although Eq. (11) does not include observations above 8 km, the area bounded between $f(z)_{75th}$
and $f(z)_{25th}$ in higher altitudes is too small to make a noticeable impact on this adjustment factor.
One may object that since we estimated the adjustment factor based on two boundaries
($25^{th}$ and $75^{th}$ percentiles) of the data we are no longer really dealing with 50% of features observed
in the vertical shapes of FNR. This valid critique can be overcome by gradually relaxing the lower
and upper limits and examine the resulting change in $f_{\text{adj}}$. When we reduce the lower limit in Eq.
(9) from the $25^{th}$ to $1^{st}$ percentiles the optimal curve is similar to the one shown in Figure 6 (Figure
S7). However, when we extend the upper limit from $75^{th}$ percentiles to greater values, we see the
fit becoming less robust above the $80^{th}$ percentiles indicating that the formulation is applicable for
~80% of the data. The reason behind the poor representation of the adjustment factor for the upper
tail of the population is the extremely steep turning point between 5.5 and 6.0 km necessitating a
higher order rational function to be used for Eq. (7) and Eq. (8). We prefer to limit this analysis to
both boundaries and the order defined in Eq. (8) and Eq. (9) because extreme value predictions
usually lack robustness.
A caution with these results is that our analysis is limited to afternoon observations because
our focus is on afternoon low orbiting sensors such as OMI and TROPOMI. Nonetheless,
Schroeder et al. (2017) and Crawford et al. (2021) observed a large diurnal variability in these
profiles due to diurnal variability in  sinks and sources of $NO_2$ and HCHO, and atmospheric
dynamics. The diurnal cycle has indeed an important implication for geostationary satellites such
as Tropospheric Emissions: Monitoring of Pollution (TEMPO) (Chance et al., 2019).
Another important caveat with our analysis is that it is based upon four air quality
campaigns taking place in warm seasons avoiding times/areas with convective transport; as such
our analysis is ignorant about the vertical shapes of FNR during convective activities and cold
seasons. These oversights can be downplayed by a few compelling assumptions: first, it is very
atypical to encounter elevated ozone production rates during cold seasons with few exceptions
(Ahmadov et al., 2015; Rappenglück et al., 2014); second, the notion of ozone regimes is only
appropriate in photochemically active environments where the ROx-HOx cycle is active; an
example of this can be found in Souri et al. (2021) who observed an enhancement of surface ozone
in central Europe during a lockdown in April 2020 (up to 5 ppbv) compared to a baseline which
was explainable by the reduced $O_3$ titration through NO in place of the photochemically induced
production. An exaggerated extension to this example is the nighttime chemistry where NO-$O_3$-
$NO_2$ partitioning is the major driver of negative ozone production rates; at night, the definition of
NOx-sensitive or VOC-sensitive is meaningless, so is in photochemically less active
environments; third, it is rarely advisable to use cloudy scenes in satellite UV-Vis gas retrievals
due to the arguable assumption on Lambertian clouds and highly uncertain cloud optical centroid
and albedo; accordingly, convection occurring during storms or fires are commonly masked in
satellite-based studies. Therefore, the limitations associated with the adjustment factor are mild
compared to the advantages.

### 3.6. *Spatial Heterogeneity*


The spatial representation error resulting from both unresolved processes and scales (Janić
et al., 2016; Valin et al., 2011; Souri et al., 2022) refers to the procedure of the quantification of
the amount of information lost due to satellite footprint or unresolved inputs used in satellite
retrieval algorithms. This source of error cannot be determined when we do not know the true state
of the spatial variability. There is, however, a practical way to determine this by conducting multi-
scale intercomparisons of a coarse spatial resolution output against a finer one. Yet, despite the
absence of the truth in this approach, we tend to find their comparisons useful in giving us an
appreciation of the error.
We build the reference data on qualified pixels (qa_value> 0.75) of offline TROPOMI
tropospheric $NO_2$ version 2.2.0 (van Geffen et al., 2021; Boersma et al., 2018) and total HCHO
columns version 2.02.01 (De Smedt et al., 2018) oversampled at 3×3 $km^2$ in summer 2021 over
the US. Figure 7 shows the map of those tropospheric columns as well as FNR. Encouragingly,
the small footprint and relatively low detection limit of TROPOMI compared to its predecessor
satellite sensors (e.g., OMI) enables us to have possibly one of the finest maps of HCHO over the
US to date. Large values of HCHO columns are found in the southeast due to strong isoprene
emissions (e.g., Zhu et al., 2016; Wells et al., 2020). Cities like Houston (Boeke et al., 2011; Zhu
et al., 2014; Pan et al., 2015; Diao et al., 2016), Kansas City, Phoenix (Nunnermacker et al., 2004),
and Los Angeles (de Gouw et al., 2018) also show pronounced enhancements of HCHO possibly
due to anthropogenic sources. Expectedly, large tropospheric $NO_2$ columns are often confined to
cities and some coal-fired power plants along Ohio river basin. Concerning FNR, low values
dominate cities whereas high values are found in remote regions. An immediate tendency observed
from these maps is that the length scale of HCHO columns is longer than that of $NO_2$. This
indicates that $NO_2$ columns are more heterogenous. It is because of this reason that we observe a
large degree of the spatial heterogeneity with respect to FNRs.
Here we limit our analysis to Los Angeles due to computational costs imposed by the
subsequent experiment. To quantify the spatial representation errors caused by satellite footprint
size, we upscale the FNRs by convolving the values with four low pass box filters with the size of
13×24, 36×36, 108×108, and 216×216 $km^2$, shown in the first column of Figure 8. Subsequently,
to extract the spatial variance (information), we follow the definition of the experimental
semivariogram (Matheron, 1963):

$$\gamma(\boldsymbol{h}) = \frac{1}{2N(\boldsymbol{h})} \sum_{|x_i-x_j|-|\boldsymbol{h}|\leq\varepsilon} [Z(x_i) - Z(x_j)]^2 \qquad (12)$$

where $Z(x_i)$ (and $Z(x_j)$) is discrete pixels of FNRs, $N(\boldsymbol{h})$ is the number of paired pixels separated
by the vector of $\boldsymbol{h}$. |.| operator indicates the length of a vector. The condition of $|x_i - x_j| - |\boldsymbol{h}| \leq$
$\varepsilon$ is to permit certain tolerance for differences in the length of the vector. Here, we rule out the
directional dependency in $\gamma(\boldsymbol{h})$, which in turn, makes the vector of $\boldsymbol{h}$ scalar ($h = |\boldsymbol{h}|$). Moreover,
we bin $\gamma$ values in 100 evenly-spaced intervals ranging from 0 to 5 degree. To model the
semivariogram, we follow the stable Gaussian function used in Souri et al. (2022):

$$\gamma(h) = s(1 - e^{-(\frac{h}{r})^{c_0}}): c_0=1.5 \qquad (13)$$

where $r$ and $s$ are fitting parameters. For the most part, geophysical quantities become spatially
uncorrelated at a certain distance called the range and the variance associated with that distance is
called the sill. that the fitting parameters $r$ and $s$ describe these two quantities as long as the stable





Gaussian function can well fit to the shape of semivariogram. The semivariograms, and the fits,
associated with each map is depicted in the second column of Figure 8.
The modeled semivariograms suggest that a coarser field comes with a smaller sill,
implying a loss in the spatial information (variance). The length scale (i.e., the range) only sharply
increases at coarser footprints (>36×36 km$^2$). This indicates that several coarse resolution satellite
sensors such as OMI (13×24 km$^2$) are rather able to determine the length scales of FNR over a
major city such as Los Angeles. By leveraging the modeled semivariograms, we can effortlessly
determine the spatial representation error for specific scale (e.g., $h$=10 km) through

$$e^2(h) = 1 - \frac{\gamma(h)}{\gamma_{ref}(h)} \tag{14}$$

where $\gamma_{ref}(h)$ is the modeled semivariogram of the reference field (3×3 km$^2$). Figure 9 shows the
spatial representation errors for different fields and different length scales. For the most part, the
OMI nadir pixel (13×24 km$^2$) only have a ~12% loss of the spatial variance. On the contrary, a
grid box with a size of 216×216 km$^2$ fails at capturing ~65% of the spatial information in FNR
with a 50 km length scale comparable to the extent of Los Angeles. The advantage of our method
is that we can mathematically describe the spatial representation error as function of the length of
our target. The present method can be easily applied to other atmospheric compounds and
locations. We have named this method SpaTial Representation Error EstimaTor (STREET) which
is publicly available as an open-source package ([https://github.com/ahsouri/STREET](https://github.com/ahsouri/STREET), Last
Access, May 31, 2022).
An oversight in the above experiment lies in its lack of appreciation of unresolved physical
processes in the satellite measurements: weaker sensitivity of some spectra windows to the near-
surface pollution (Yang et al., 2014), using 1-D air mass factor calculation instead of 3-D
(Schwaerzel et al., 2020), and discounting aerosol effect on the light path are just few examples to
point out. To account for the unresolved processes, one can recalculate Eqs. (12)-(14) using outputs
coming from different retrieval frameworks, which is beyond the scope of this study.
### *3.7.   Satellite errors*
### *3.7.1.   Concept*
Two types of retrieval errors can affect our analysis: systematic errors (bias) and
unsystematic ones (random errors). In theory, it is very compelling to understand their differences.
In reality, the distinction between random and systematic errors is not as clear-cut as it seems. One
may wish to establish the credibility of a satellite retrieval by comparing it to a sky-radiance
measurement over time. Because each measurement is made at a different time, their comparison
is not a repetition of the same experiment; each time, the atmosphere differs in some aspects so
each comparison is unique. Adding more sky-radiance measurements will simply add new
experiments. For each paired data points, there are many unique issues contributing differently to
errors; as such our problem is grossly under-determined (i.e., more unknowns for a given
observation). Here, we do not attempt to separate those types of errors in the subsequent analysis,
thereby limiting the analysis to the total uncertainty.
We focus on analyzing the statistical errors drawn from the differences between benchmark
and the retrievals on daily basis. Two sensors are used for this analysis: TROPOMI and OMI. To
propagate individual uncertainties in HCHO and NO$_2$ to FNRs, we follow an analytical approach
involving Jacobians of the ratio to HCHO and NO$_2$. Assuming that errors in HCHO and NO$_2$ are
uncorrelated, the relative error of the ratio can be estimated by:





$$\frac{\sigma}{ratio} = \sqrt{\left(\frac{\sigma_{HCHO}}{HCHO}\right)^2 + \left(\frac{\sigma_{NO_2}}{NO_2}\right)^2} \tag{15}$$

where $\sigma_{HCHO}$ and $\sigma_{NO_2}$ are total uncertainties of HCHO and $NO_2$ observations.
*3.7.2. Error Distributions in TROPOMI and OMI*
We begin our analysis with the error distribution of daily TROPOMI tropospheric $NO_2$
columns (v1.02.02) against 22 MAX-DOAS instruments from May to Sep in 2018-2021. The data
are paired based on the criteria defined in Verhoelst et al. (2021). The list of stations and the
number of available days for each are mapped in Figure S8. Figure 10a shows the histogram of the
TROPOMI minus the MAX-DOAS instruments. The first observation from this distribution is that
it is skewed towards lower differences evident in the skewness parameter around -4.6. As a result
of the skewness, the median should be a better representative of the central tendency which is
around $-1\times10^{15}$ molec./cm$^2$. In general, TROPOMI tropospheric $NO_2$ columns show a low bias.
We fit a normal distribution to the data using non-linear Levenberg-Marquardt method. This fitted
normal distribution (R$^2$=0.94) is used to approximate $\sigma_{NO_2}$ for different confidence intervals and
to play down blunders.
The error analysis for OMI follows the same methods applied for TROPOMI; however,
with different benchmarks. We follow the comparisons made between the operational product
version 3.1 and measured columns derived from NCAR's $NO_2$ measurements integrated along
aircraft spirals during four NASA's air quality campaigns. More information regarding this data
comparison can be found in Choi et al. (2020). Figure 10b shows the histogram of OMI minus the
integrated spirals. Compared to TROPOMI, the OMI bias is worse by a factor of two. The standard
deviation calculated from a Gaussian fit ($2.31\times10^{15}$ molec./cm$^2$) is not substantially different than
that of TROPOMI ($2.11\times10^{15}$ molec./cm$^2$).
As for the error distribution of TROPOMI HCHO columns (version 1.1.(5-7)), we use 24
FTIR measurements during the same time period based on the criteria specified in Vigouroux et
al. (2020). The stations along with their number of available data are mapped in Figure S8. The
frequency of the paired data is daily. Figure 11a depicts the error distribution. The distribution is
slightly broader compared to that of $NO_2$, manifested in a larger standard deviation $4.32\times10^{15}$
molec./cm$^2$. This is primarily due to the fact that the molecular absorption of HCHO is much
smaller/narrower than that of $NO_2$ in the UV-Vis range (Gonzalez Abad et al., 2019);
consequently, HCHO observations are more contaminated by noise. Similar to the $NO_2$, we fit a
normal distribution (R$^2$=0.90) to specify $\sigma_{HCHO}$ for different confidence intervals.
Concerning OMI HCHO columns from SAO version 3 (Gonzalez Abad et al., 2015), we
follow the intercomparison approach proposed in Zhu et al. (2020). Based on this approach, the
benchmarks come from GEOS-Chem simulated HCHO columns corrected by in-situ aircraft
measurements. The measurements were made during ozone seasons from KORUS-AQ,
DISCOVERs, FRAPPE, NOMADSS, and SENEX campaigns (see Table 1 in Zhu et al. 2020).
OMI values ranging from $-0.5\times10^{15}$ molec./cm$^2$ and $1.0\times10^{17}$ molec./cm$^2$ with effective cloud
fraction between 0.0 and 0.3, and SZA between 0 and 60 degrees are only considered in the
comparison. Any pixels from OMI and grid boxes from the corrected GEOS-Chem simulation that
fall into a polygon enclosing the campaign domain are used to create the error distribution shown
in Figure 11b. The distribution has much denser data because the model output covers a large
portion of the satellite swath. The error distribution suggests that OMI HCHO is inferior to
TROPOMI evident in larger bias and standard deviation. The OMI bias is twice as large as that of



TROPOMI. De Smedt et al. (2021) observed the same level of bias from their comparisons of
OMI/TROPOMI with MAX-DOAS instruments (see Table 3 in their paper). Moreover, their OMI
vs MAX-DOAS comparisons were severely scattered. Likewise, we observe the standard deviation
of OMI from the fitted Gaussian function to be roughly five times as large of that TROPOMI. This
can primarily due to a weaker signal-to-noise (and sensor degradation) in OMI. It is because of
this reason that OMI HCHO should be oversampled for few months. Another possible reason for
the large standard deviation is the fact that the benchmark arises from a modeling experiment
whose ability at resolving spatiotemporal variations in HCHO may be uncertain. This partly leads
to the performance of OMI to look poor.

### 3.7.3. The impact of retrieval error on the ratio

Following Eq. (15), we calculate the standard error for a wide range of $NO_2$ and HCHO
columns at three confidence intervals based on the standard deviation derived from the fitted
Gaussian function to the histograms. The resultant standard error (%) at 68% confidence interval
(1 sigma) for both TROPOMI and OMI is shown in Figure 12. We observe smaller errors to be
associated with larger tropospheric column concentrations. As for TROPOMI, either daily HCHO
or tropospheric $NO_2$ columns should be above $1.2\text{-}1.5 \times 10^{16}$ molec./cm$^2$ to achieve 20-30%
standard error. The TROPOMI errors start diminishing the application of FNR when both
measurements are below this threshold. Regarding OMI, it is nearly impossible to get the standard
error below of 20-30% given its problematically large HCHO standard deviation. For 50% error,
the daily HCHO columns should be above $3.2 \times 10^{16}$ molec./cm$^2$. This range of error can also be
achieved if OMI tropospheric $NO_2$ columns are above $8 \times 10^{15}$ molec./cm$^2$.

### 3.8. The fractional errors to the total error

The ultimate task is to compile the aforementioned errors to gauge how each individual
source of error contributes to the overall error. The overall error is given by:

$$\sigma_{total} = \sqrt{\sigma_{Col2PBL}^2 + \sigma_{SpatialRep}^2 + \sigma_{Retreival}^2} \tag{16}$$

$\sigma_{Col2PBL}$ is the error in the adjustment-factor defined in this study. We calculated a 26% standard
error for a wide range of PBLHs. Therefore, $\sigma_{Col2PBL}$ equals to 26% of the observed ratio (i.e.,
magnitude dependent). $\sigma_{SpatialRep}$ is more complex. It is a function of the footprint of the satellite
(or a model), the spatial variability of the reference field which varies from environment to
environment, and the length scale of our target (e.g., a district, a city, or a state). Eq. (14) explicitly
quantifies this error. The product of the square root of that value and the observed ratio defines
$\sigma_{SpatialRep}$. The last error depends on the magnitude of HCHO and $NO_2$ tropospheric columns. It
can be estimated from Eq. (15) times the observed ratio. We did not include the chemistry error in
Eq. (16) because it was suited only for segregating the chemical conditions; it does not describe
the level of uncertainties that comes with the observed columnar ratio. Figure 13 shows the total
relative error given the observed TROPOMI ratio seen in Figure 7. We consider the OMI spatial
representation error (13% variance loss) for this case that was computed in a city environment.
The retrieval errors are based on TROPOMI sigma values. Areas associated with relatively small
errors (<50%) are mostly seen in cities due to a stronger signal (smaller $\sigma_{Retreival}$). Places with
low vegetation and anthropogenic sources (i.e., Rocky Mountains) possess the largest errors
(>100%).
To produce some examples of the fractional errors to the total error, we focus on two
different environments with two different sets of HCHO and $NO_2$ columns. One represents a



heavily polluted area, and the other one a moderately polluted region. We also include two
footprints: OMI ($13 \times 24$ km$^2$) and a $108 \times 108$ km$^2$ pixel. Finally, we calculate the percentage of
each error component for both OMI and TROPOMI sensors. Figure 14 shows the pie charts
describing the percentage of each individual error to the total error for TROPOMI. Unless the
footprint of the sensor is coarse enough (e.g., 108 km$^2$) to give rise to the spatial representation
error dominance, the retrieval error stands out. It is not expected for new satellites to have very
large footprints; as such, the retrieval errors appear to be the major obstacle for using FNR in a
robust manner. Figure 15 shows the same calculation but using OMI errors; the retrieval errors
massively surpass other errors. This motivates us to do one more experiment; we recalculate the
HCHO error distribution in OMI using monthly-averaged data instead of daily (Figure S9). This
experiment suggests a standard deviation of $9.4 \times 10^{15}$ molec./cm$^2$ with which we again observe
the retrieval error to be the largest contributor (>80%) of the total error (Figure S10).
**4.   Summary**
The main goal of this study was to characterize the errors associated with the ratio of
satellite-based HCHO to NO$_2$ columns which has been widely used for ozone sensitivity studies.
From the realization of the complexity of the problem we now know that four major errors should
be carefully quantified so that we can reliably represent the underlying ozone regimes. The errors
are broken down into i) the chemistry error, ii) the column to the PBL translation, iii) the spatial
representation error, and iv) the retrieval error. Each error has its own dynamics and has been
tackled differently by leveraging a broad spectrum of tools and data.
The chemistry error refers to the predictive power of HCHO/NO$_2$ ratio (hereafter FNR) at
describing the HOx-ROx cycle which can be well explained by the ratio of the chemical loss of
HO$_2$+RO$_2$ (LROx) to the chemical loss of NOx (LNOx). Because those chemical reactions are not
directly observable, we set up a chemical box model constrained with a large suite of in-situ aircraft
measurements collected during DISCOVER-AQs and KORUS-AQ campaigns (~ 500 hr of flight).
Our box model showed a reasonable performance at recreating some of unconstrained key
compounds such as OH (R$^2$=0.64, bias=17%), HO$_2$ (R$^2$=0.66, bias<1%), and HCHO (R$^2$=0.73,
bias=5%). Subsequently we compared the simulated FNRs to LROx/LNOx. They showed a high
degree of correspondence (R$^2$=0.93) but only in the log-log scale; this indicated that FNRs poorly
described the HOx-ROx cycle for heavily polluted environments as well as pristine ones.
Following a robust baseline indicator (ln(LROx/LNOx) = -1.0 ± 0.2) segregating NOx-sensitive
from VOC-sensitive regimes, we observed a diverse range of FNR ranging from 1 to 4. These
transitioning ratios had a Gaussian distribution with a mean of 1.8 and standard deviation of 0.4.
This implied that the relative standard error associated with the ratio from the chemistry
perspective at 68% confidence interval was 20%. Although this threshold with its error was based
on a single model realization and can be different for a different chemical mechanism, it provided
a useful universal baseline derived from various chemical and meteorological conditions. At 68%
confidence level, any uncertainty beyond 20% in the ozone regime identification from FNRs likely
originates from other sources of error such as the retrieval error.
Results from the box model showed that ozone production rates in extremely polluted
regions (VOC-sensitive) were not significantly different than those in pristine ones (NOx-
sensitive) due to non-linear chemical feedback mostly imposed by NO$_2$+OH. Indeed, the largest
PO$_3$ rates (median = 4.6 ppbv/hr) were predominantly seen in VOC-sensitive regimes tending
towards the transitional regime. This was primary caused by the abundance of ozone precursors
(i.e., HCHO×NO$_2$) in addition to the diminished negative chemical feedback. We also revealed
that HCHO×NO$_2$ can be used as a sensible proxy for the ozone precursors abundance. In theory,





this metric in conjunction with the ratio provided reasonable estimates on $PO_3$ rates ($\pm 0.60$
ppbv/hr).
We then analyzed the afternoon vertical distribution of HCHO, $NO_2$, and their ratio
observed from aircrafts during the air quality campaigns binned to the near surface to 8 km. For
altitudes below 5.75 km, HCHO concentration steadily decreased with altitude but at a smaller rate
compared to $NO_2$. Above that altitude, $NO_2$ concentrations stabilized and slightly increased due to
lightning and stratospheric sources. The dissimilarity between the vertical shape of $NO_2$ versus
HCHO resulted in a non-linear shape of FNR. This non-linear shape necessitated a mathematical
formulation to transform an observed columnar ratio to a ratio at a desired vertical height
expanding from the surface. We fit a second-order rational function to the profile and formulated
the altitude adjustment factor which clearly followed a second-order polynomial function starting
from values below 1 for lower altitudes, following values above 1 for some high altitudes, and
finally converging to 1 at 8 km. This behavior means that for a given tropospheric columnar ratio,
the ozone regime tends to get pushed towards the VOC-sensitive regime near the surface. This
data-driven adjustment factor exclusively derived from afternoon aircraft profiles during warm
seasons in non-convective conditions had a standard error of 26%.
An important error in the satellite-based observations stemmed from unresolved spatial
variability in trace gas concentrations within a satellite pixel (Souri et al., 2022; Tang et al., 2021).
The amount of unresolved spatial variability (the spatial representation error) can in principle be
modeled if we base our reference on a distribution map made from a high spatial resolution dataset.
We modeled semivariograms (or spatial auto-correlation) computed for a reference map of FNR
observed by TROPOMI at $3\times3$ km$^2$ over Los Angeles. Subsequently, we coarsened the map to
$13\times24$, $36\times36$, $108\times108$, and $216\times216$ km$^2$ and modeled their semivariograms. As for $13\times24$ km$^2$,
which is equivalent of the OMI nadir spatial resolution, around 12% of spatial information
(variance) was lost due to its footprint. The larger the footprint, the bigger spatial representation
error. For instance, a grid box with the size of $216\times216$ km$^2$ lost 65% of the spatial information in
the ratio at 50 km length scale. Our method is compelling to understand and easy to apply for other
products and different atmospheric environments. We developed an open-source package called
SpaTial Representation Error EstimaTor (STREET) (https://github.com/ahsouri/STREET) based
on this approach.
We presented estimates of retrieval errors associated with daily TROPOMI and OMI
tropospheric $NO_2$ columns by comparing them against a large suite of MAX-DOAS (Verhoelst et
al. 2021) and vertically-integrated measurements from aircraft spirals (Choi et al., 2020). Both
products were smaller than the benchmark. Furthermore, they show a relatively consistent
dispersion at 68% confidence level ($\sim2\times10^{15}$ molec./cm$^2$) suggested by fitting a normal function
($R^2>0.9$) to their error distributions. As for daily TROPOMI and OMI HCHO products, we used
global FTIR observations (Vigouroux et al., 2020) and data-constrained GEOS-Chem outputs from
multiple campaigns (Zhu, 2020), respectively. TROPOMI HCHO indeed outperforms OMI
HCHO with respect to bias and dispersion on a daily basis. The standard deviation of OMI HCHO
was found to be roughly five times as large compared to TROPOMI. While this error can be partly
reduced by oversampling over a span of a month or a season, it is critical to recognize that ozone
events are episodic, thus daily observations should be the standard mean for understanding the
chemical pathways for the formation of surface ozone. After combining the daily biases from both
HCHO and $NO_2$ TROPOMI comparisons, we came to the conclusion that either daily HCHO or
tropospheric $NO_2$ columns should be above $1.2$-$1.5\times10^{16}$ molec./cm$^2$ to achieve 20-30% standard
error in the ratio. Due to the large error in daily OMI HCHO, it was nearly impossible to achieve



20-30% standard error given the observable range of HCHO and $NO_2$ columns over our planet. To
reach to 50% error using daily OMI data, either HCHO columns should be above $3.2×10^{16}$
molec./cm$^2$ or tropospheric $NO_2$ columns should be above $8×10^{15}$ molec./cm$^2$.
We finally calculated the total error in the ratio by combining the TROPOMI retrieval
errors, the spatial representation error pertaining to OMI nadir footprint over a city-like
environment, and the altitude adjustment error for a wide range of observed HCHO and $NO_2$
columns over the US. These observations were based on the TROPOMI in summertime 2021. The
total errors were relatively mild (<50%) in cities due to a stronger signal, whereas they easily
exceeded 100% in regions with low vegetation and anthropogenic sources (i.e., Rocky Mountains).
The dominant source of the total error (40-90%) was the retrieval error.
All of these aspects highlight the necessity of improving the trace gas satellite retrieval
algorithms in conjunction with sensor calibration, although with the realization that a better
retrieval is somewhat limited by the advancements made in other disciplines such as atmospheric
modeling and molecular spectroscopy.
**Acknowledgment**
This study was funded by NASA's Aura Science Team (grant number: 80NSSC21K1333). PTR-
MS measurements were supported by the Austrian Federal Ministry for Transport, Innovation and
Technology (bmvit, FFG-ALR-ASAP). The PTR-MS instrument team (P. Eichler, L. Kaser, T.
Mikoviny, M. Müller) is acknowledged for their support with field work and data processing. We
acknowledge FTIR HCHO measurements team including T. Blumenstock, M. Grutter, J. W.
Hannigan, N. Jones, R. Kivi, E. Lutsch, E. Mahieu, M. Makarova, I. Morino, I. Murata, T.
Nagahama, J. Notholt, I. Ortega, M. Palm, A. Röhling, M. Schneider, D. Smale, W. Stremme, K.
Strong, Y. Sun,  R. Sussmann, Y. Té, and P. Wang. The measurements at Paramaribo have been
supported by the BMBF (German Ministry of Education and Research) in the project ROMIC-II
subproject TroStra (01LG1904A). We thank the Meteorological Service Suriname and Cornelis
Becker for support. The measurements and data analysis at Bremen are supported by the Senate of
Bremen. The NCAR FTS observation programs at Thule, GR, Boulder, CO and Mauna Loa, HI
are supported under contract by the National Aeronautics and Space Administration (NASA). The
National Center for Atmospheric Research is sponsored by the National Science Foundation. The
Thule effort is also supported by the NSF Office of Polar Programs (OPP). Operations at the
Rikubetsu and Tsukuba FTIR sites are supported in part by the GOSAT series project. The Paris
TCCON site has received funding from Sorbonne Université, the French research center CNRS
and the French space agency CNES. The Jungfraujoch FTIR data are primarily available thanks to
the support provided by the F.R.S. - FNRS (Brussels), the GAW-CH program of MeteoSwiss
(Zürich) and the HFSJG.ch Foundation (Bern). The MAX-DOAS data used in this publication
were obtained from A. Bais, J. Burrows, K. Chan, M. Grutter, C. Liu, H. Irie, V. Kumar, Y.
Kanaya, A. Piters, C. Rivera-Cárdenas, M. Van Roozendael, R. Ryan, V. Sinha, and T. Wagner.
Fast delivery of MAX-DOAS data tailored to the S5P validation was organized through the S5PVT
AO project NIDFORVAL. IUP-Bremen ground-based measurements are funded by DLR-Bonn
received through project 50EE1709A. We thank the IISER Mohali atmospheric chemistry facility
for supporting the MAX-DOAS measurements at Mohali, India. KNMI ground-based
measurements in De Bilt and Cabauw are partly supported by the Ruisdael Observatory project,
Dutch Research Council (NWO) contract 184.034.015, by the Netherlands Space Office (NSO)
for Sentinel-5p/TROPOMI validation, and by ESA via the EU CAMS-27 project. LZ and SS
acknowledge grants from Guangdong Basic and Applied Basic Research Foundation
(2021A1515110713) and Shenzhen Science and Technology Program



(JCYJ20210324104604012). The TROPOMI validation work was supported by BELSPO/ESA
through the ProDEx project TROVA-E2 (grant no. PEA 4000116692). TV acknowledges support
from BELSPO through BRAIN-BE 2.0 project LEGO-BEL-AQ (contract B2/191/P1/LEGO-
BEL-AQ). We thank Glenn Diskin for providing CO, $CO_2$, and $CH_4$ measurements. We thank Paul
Wennberg for $H_2O_2$ and $HNO_3$ measurements.

**Data Access**
The FTIR and MAXDOAS data used in this publication were partly obtained from the Network
for the Detection of Atmospheric Composition Change (NDACC) and are available through the
NDACC website www.ndacc.org. The spatial representation error is estimated based on publicly
available package, SpaTial Representation Error EstimaTor (STREET)
(https://github.com/ahsouri/STREET). DISCOVER-AQ and KORUS-AQ aircraft data can be
downloaded from https://www-air.larc.nasa.gov/missions/discover-aq/discover-aq.html and
https://www-air.larc.nasa.gov/missions/korus-aq/. TROPOMI $NO_2$ and HCHO data can be
downloaded from https://disc.gsfc.nasa.gov/datasets/S5P_L2__NO2____1/summary and
https://disc.gsfc.nasa.gov/datasets/S5P_L2__HCHO___1/summary. The box model results can be
obtained by contacting the corresponding author through ahsouri@cfa.harvard.edu.

**Author contributions**
AHS designed the research, analyzed the data, conducted the simulations, made all figures, and
wrote the paper. MSJ, SP, XL, and KC helped with conceptualization, fundraising, and analysis.
GMW helped with configuring the box model. AF, AW, WB, DRB, AJW, RCC, KM, and CC
measured various compounds during the air quality campaigns. JHC orchestrated all these
campaigns and contributed to the model interpretation. TV, SC, and GP provided paired MAX-
DOAS and TROPOMI tropospheric $NO_2$ observations. CV and BL provided paired FTIR and
TROPOMI HCHO observations. SC and LL provided paired integrated aircraft spirals and OMI
tropospheric $NO_2$ observations. LZ and SS provided the paired observations between the corrected
GEOS-Chem HCHO and OMI HCHO columns. All authors contributed to the discussion and
edited the paper.





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

9326/9/11/114004






**Table1.** The box model configurations and inputs.

| | |
|---|---|
| Temporal resolution of samples | 10-15 sec |
| Time steps | 1 hour |
| Number of solar cycles | 5 |
| Dilution constant | $1/86400$ $-1/43200$ $(s^{-1})$ |
| Meteorological Inputs | Pressure, Temperature, and Relative Humidity |
| Photolysis frequencies estimates | LUT based on the NCAR TUV model calculations |
| Photolysis frequencies constraints (campaign#‡) | Measured $jNO_2$ (1-4) and $jO^1D$ (4) |
| Compounds (Instrument#†, campaign#‡) used for constraining the box model | $H_2$(1, 4)§, CO (4, 1-4), $NO_x$ (2, 1-4), $O_3$ (2, 1-4), $SO_2$ (6, 4) , $CH_4$ (4, 1-4), $HNO_3$ (10, 1-4), Isoprene (9, 1-4), Monoterpenes (9, 1-4), Acetone (9, 1-4), Ethylene (1, 4), Ethane (1, 4), Methanol (9, 1-4), Propane (1, 4), Benzene (1 or 9, 2-4), Xylene (1 or 9, 1 and 4), Toluene (1 or 9, 1-4), Glyoxal (8, 4), Acetaldehyde (9, 1-4), Methyl vinyl ketone (9, 1-4), Methyl Ethyl Ketone (9, 2-4), Propene (1 or 9, 2 and 4), Acetic acid (9, 2-4), Glycolaldehyde (5, 4), $H_2O_2$ (5, 4) |
| Unconstrained compounds (Instrument#†, campaign#‡) used for validation | $HO_2$ (3, 4), OH (3, 4), NO (2, 1-4), $NO_2$ (2, 1-4), PAN (10, 1-4), HCHO (7, 1-4) |
| Chemical Mechanism | CB06 |

† (1) UC Irvine's Whole Air Sampler (WAS), (2) NCAR 4-Channel Chemiluminescence, (3) Penn
State's Airborne Tropospheric Hydrogen Oxides Sensor (ATHOS), (4) NASA Langley's DACOM
tunable diode laser spectrometer, (5) Caltech's single mass analyzer, (6) Georgia Tech's ionization
mass spectrometer, (7) The University of Colorado at Boulder's the Compact Atmospheric Multi-
species Spectrometer (CAMS), (8) Korean Airborne Cavity Enhances Spectrometer, (9)
University of Innsbruck's PTR-TOF-MS instrument, and (10) University of California, Berkeley's
TD-LIF.
‡ (1) DISCOVER-Baltimore-Washington, (2) DISCOVER-Texas-Houston, (3) DISCOVER-
Colorado, and (4) KORUS-AQ
§ In the absence of measurements, a default value of 550 ppbv is specified.




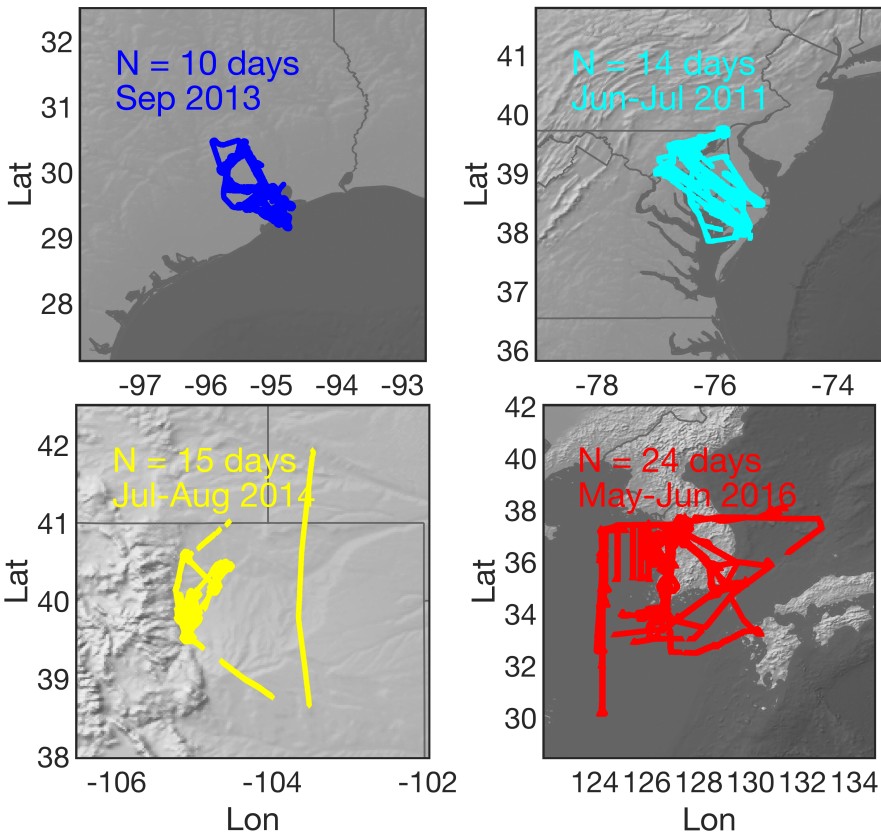

**Figure 1.** The spatial distributions of aircraft measurements collected during NASA's a)
DISCOVER-AQ Houston-Texas, b) DISCOVER-AQ Baltimore-Washington, c) DISCOVER-AQ
Colorado, and d) KORUS-AQ. The duration of each campaign is based on how long the aircraft
was in the air.

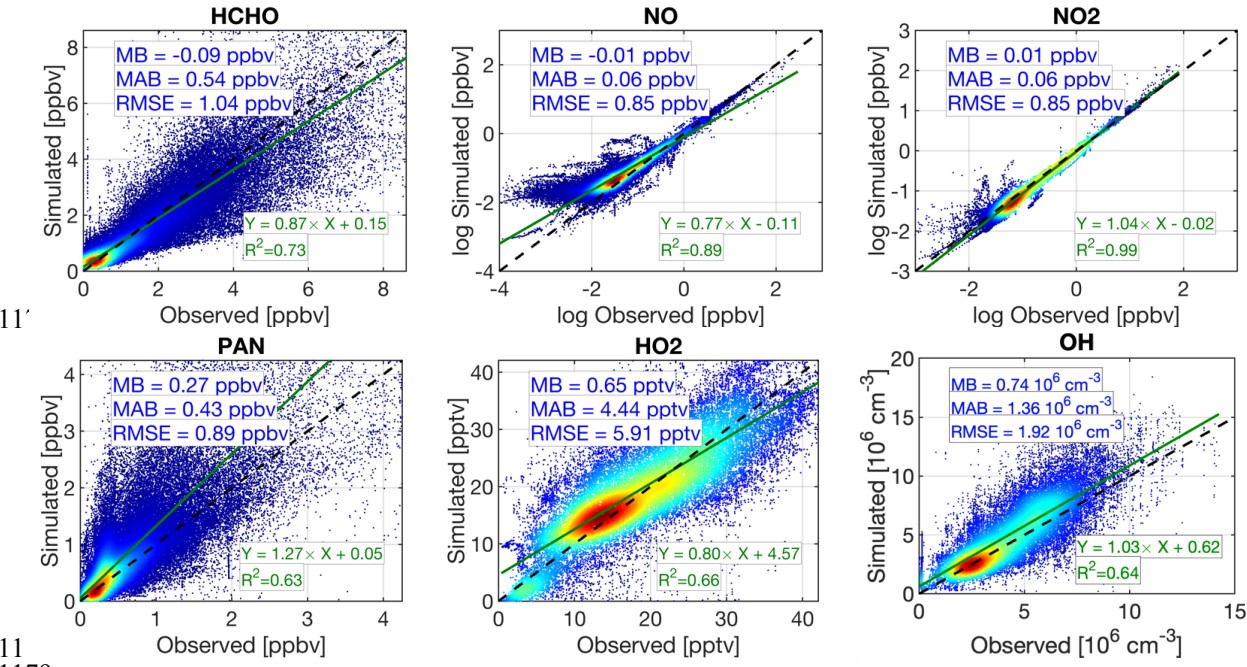

**Figure 2.** The comparisons of the observed concentrations of several critical compounds to those simulated by our F0AM box model. Each subplot contains mean bias (MB), mean absolute bias (MAB), and root mean square error (RMSE). The least-squares fit to the paired data along with the coefficient of determination ($R^2$) is also individually shown for each compound. Note that we do not account for the observations errors in the x-axis. The concentrations of NO and $NO_2$ are log-transformed.



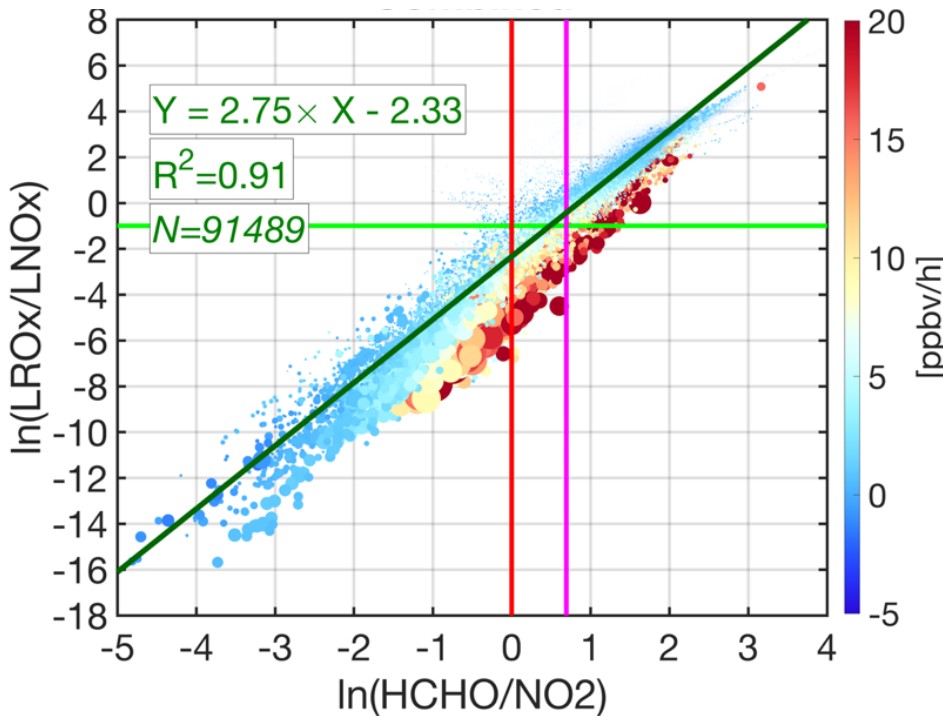

**Figure 3.** The scatterplot of natural logarithm-transformed of $HCHO/NO_2$ versus $LROx/LNOx$ based on the simulated values performed by the F0AM box model. The heat color indicates the calculated ozone production rates ($PO_3$). The size of each data point is proportional to $HCHO \times NO_2$. The light green line is the baseline separator of NOx-sensitive (above the line) and VOC-sensitive (below the line) regimes. We overlay $HCHO/NO_2 = 1$ and $HCHO/NO_2 = 2$ as red and purple lines, respectively. The dark green line indicates the least-squares fit to the paired data. The $HCHO/NO_2 = 1.8$ with 20% error is the optimal transitioning point based on this result.



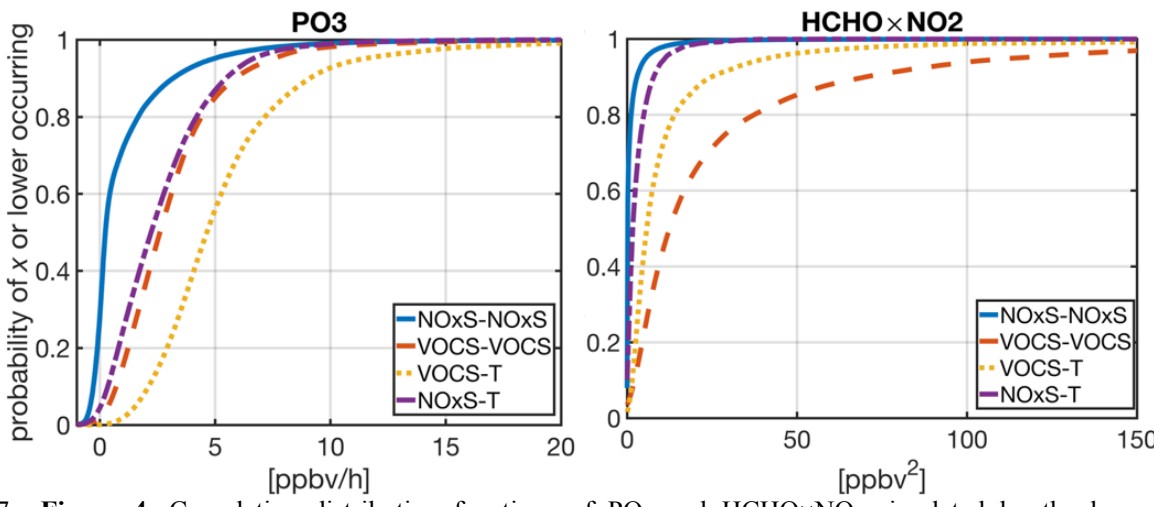

119

**Figure 4.** Cumulative distribution functions of PO$_3$ and HCHO×NO$_2$ simulated by the box model constrained by NASA's aircraft observations. Four regions namely as NOx-sensitive — NOx-sensitive, NOx-sensitive—transitional, VOC-sensitive—transitional, and VOC-sensitive—VOC-sensitive are shown. The first name of the regime is based on the baseline (ln(LROx/LNOx)=-1.0), whereas the second one follows those defined in Duncan et al. (2010): VOC-sensitive if HCHO/NO$_2$<1, transitional if 1<HCHO/NO$_2$<2, and NO$_x$-sensitive if HCHO/NO$_2$>2.

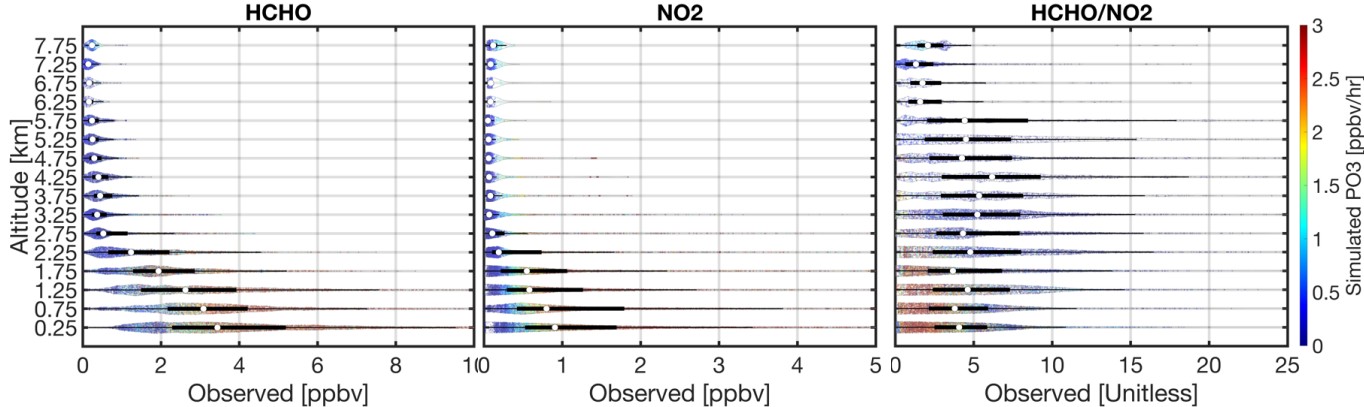

**Figure 5.** The violin plots of the afternoon vertical distrubution of HCHO, NO$_2$, and HCHO/NO$_2$
observations collected during DISCOVER-AQ Texas, Colorado, Maryland, and KORUS-AQ campaings.
The violin plots demonstrate the distrubtion of data (i.e., a wider width means a higher frequency). The
median is shown by white dots. Both 25[th] and 75[th] percentiles are shown by a solid black line. The
heatmap denotes the simulated ozone prooduction rates.

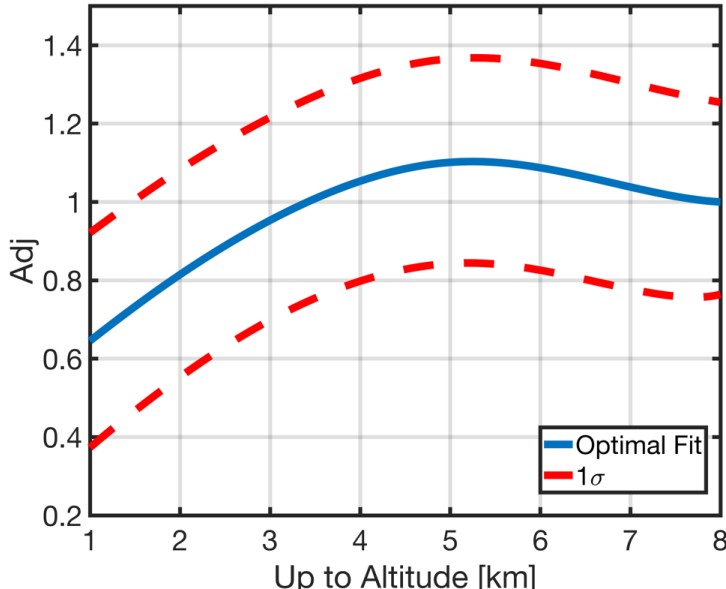

**Figure 6.** The adjustment factor defined as the ratio of the centriod (first moment) of the polygon
bounding $25^{th}$ and $75^{th}$ percentiles of the observed $HCHO/NO_2$ columns by the NASA's aircraft
between the surface to 8 km to the ones between the surface and a desired altitude. This factor can
be easily applied to the observed $HCHO/NO_2$ columns to translate the value to a desired altitude
stretching down to the surface (i.e., PBLH). The optimal curve follows a quadratic function
formulated in Eq11.
1219



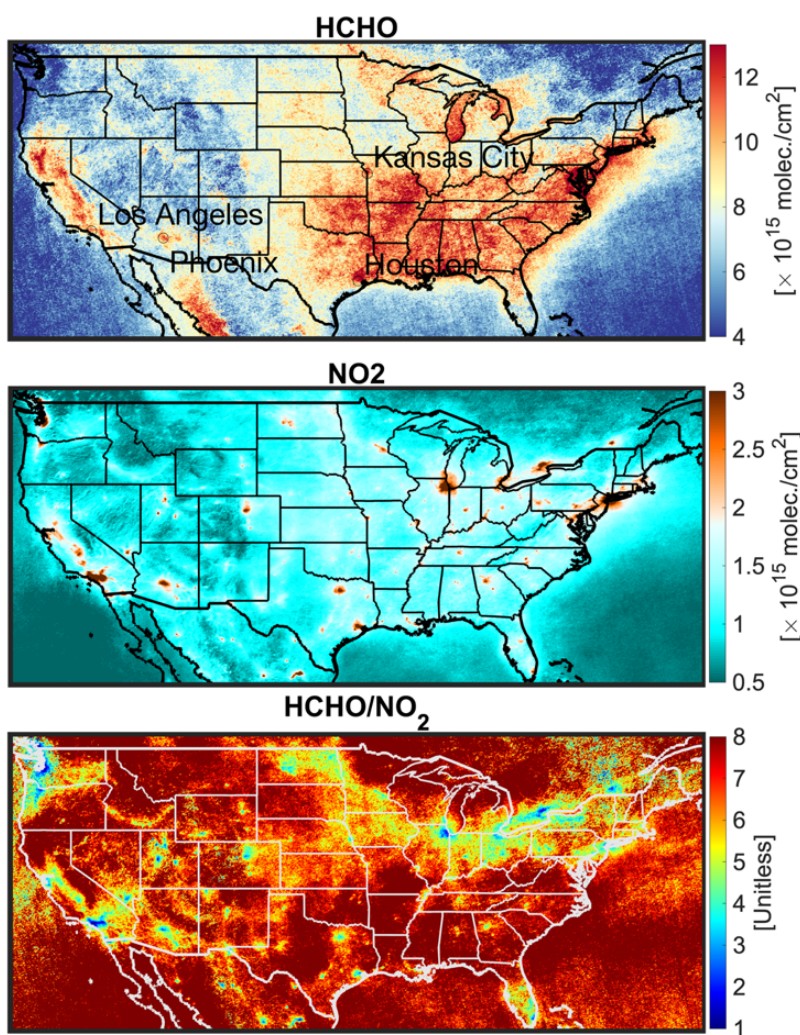

**Figure 7.** Oversampled TROPOMI total HCHO columns (top), tropospheric $NO_2$ columns (middle), and the ratio (bottom) at $3\times3$ km$^2$ from June till August 2021 over the US.

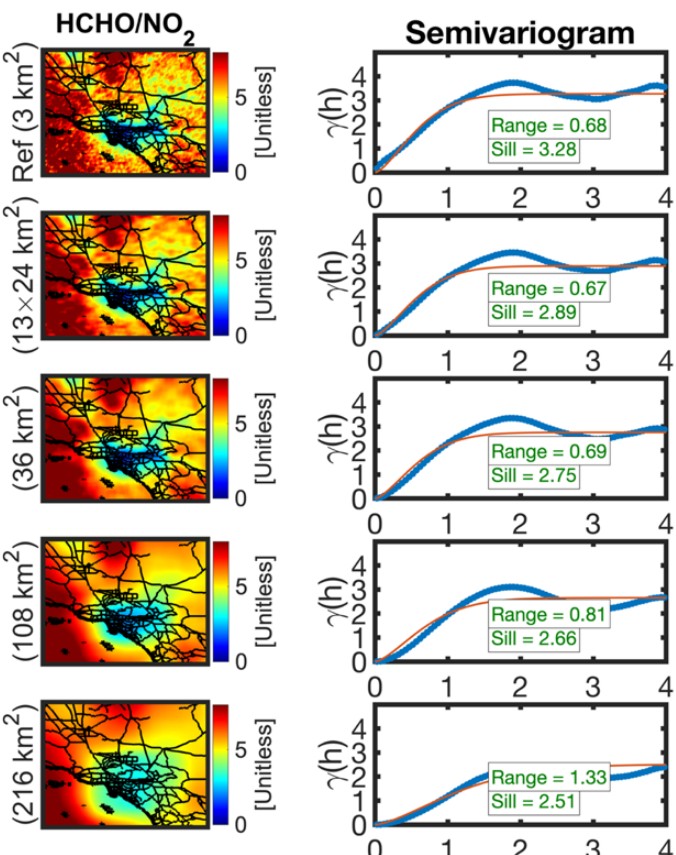

**Figure 8.** The first column represents the spatial map of HCHO/NO$_2$ ratios over Los Angeles in June till August 2021 at different spatial resolutions. To upscale each map to a coarser footprint, we use an ideal box filter tailored to the target resolution. The second column shows the semivariograms corresponding to the left map along with the fitted curve (red line). The sill and the range are computed based on the fitted curve. The x-axis in the semivariogram is in degree (1 degree ~ 110 km).

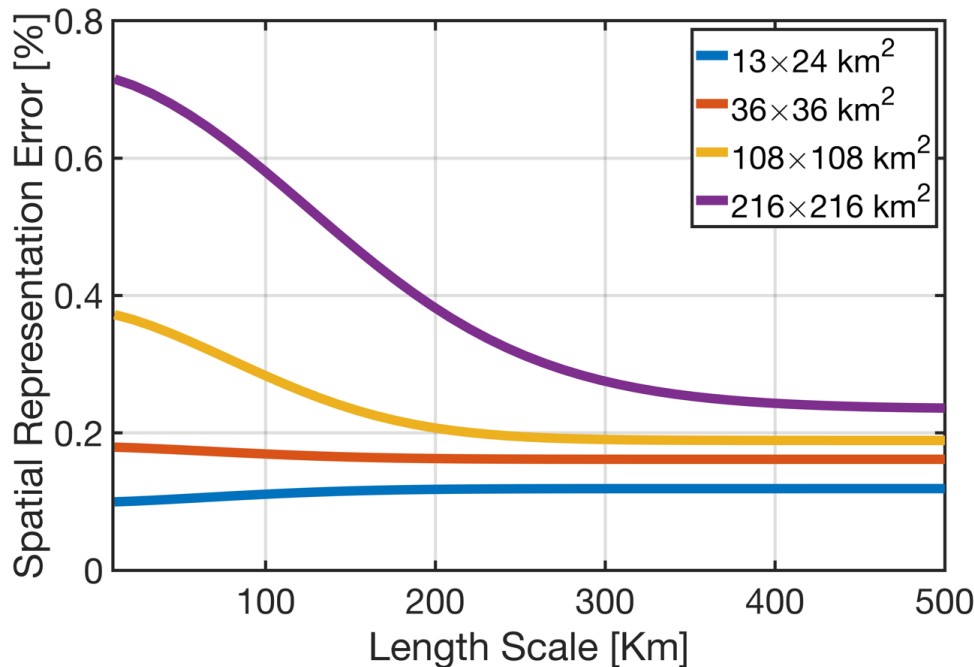

**Figure 9.** The spatial representation errors quantified based on the proposed method in this study. The error explains the spatial loss (or variance) due to the footprint of a hypothetical sensor at different length scales. To put this error in perspective, a grid box with $216 \times 216$ km$^2$ will naturally lose 65% of the spatial variance existing in the ratio at the scale of Los Angeles which roughly is 50 km wide. All of these numbers are in reference to the TROPOMI $3 \times 3$ km$^2$.



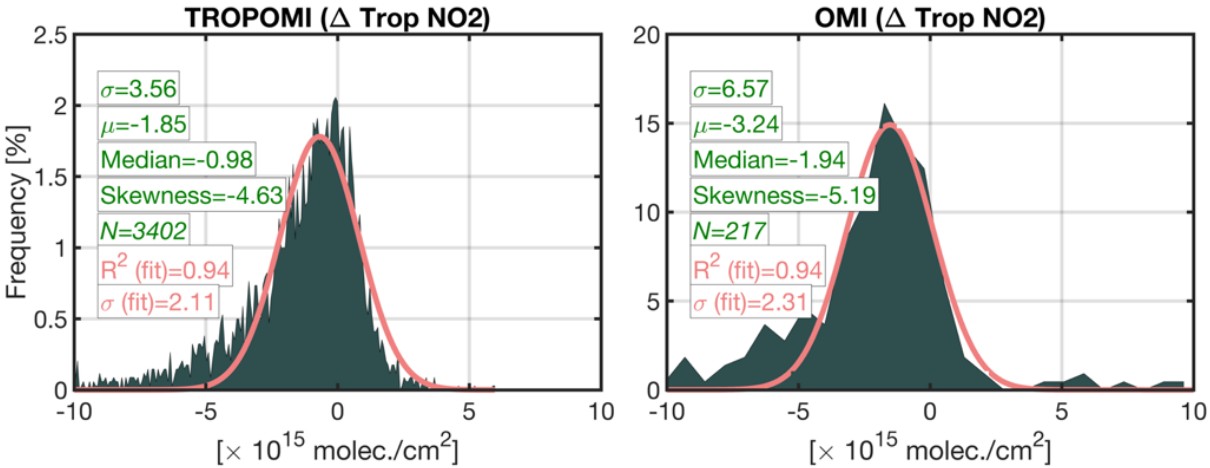

**Figure 10.** The histogram of the differences between TROPOMI and OMI and benchmarks. MAX-DOAS and integrated aircraft spirals are the TROPOMI and the OMI benchmarks, respectively. The data curation and relevant criteria on how they have been paired can be found in Verholest et al. (2021) and Choi et al. (2020). The statistics in green color are based on all data, whereas those in pink are based on the fitted Gaussian function.



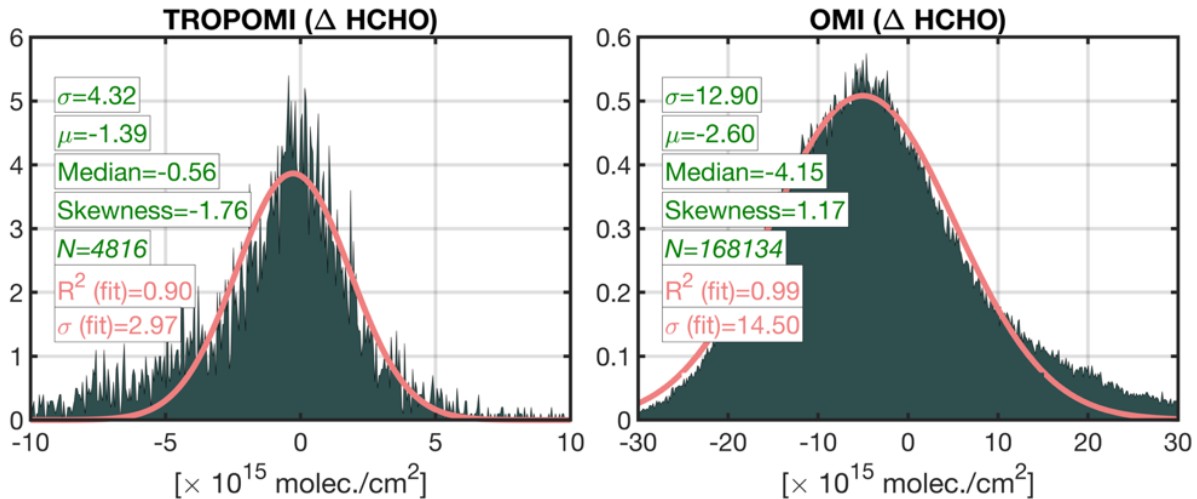

**Figure 11.** The histogram of the differences between TROPOMI and OMI and benchmarks. FTIR
and corrected GEOS-Chem simulations are respectively the TROPOMI and the OMI benchmarks.
The data curation and relevant criteria on how they have been paired can be found in Vigouroux
et al. (2021) and Zhu et al. (2020). The statistics in green color are based on all data, whereas those
in pink are based on the fitted Gaussian function.


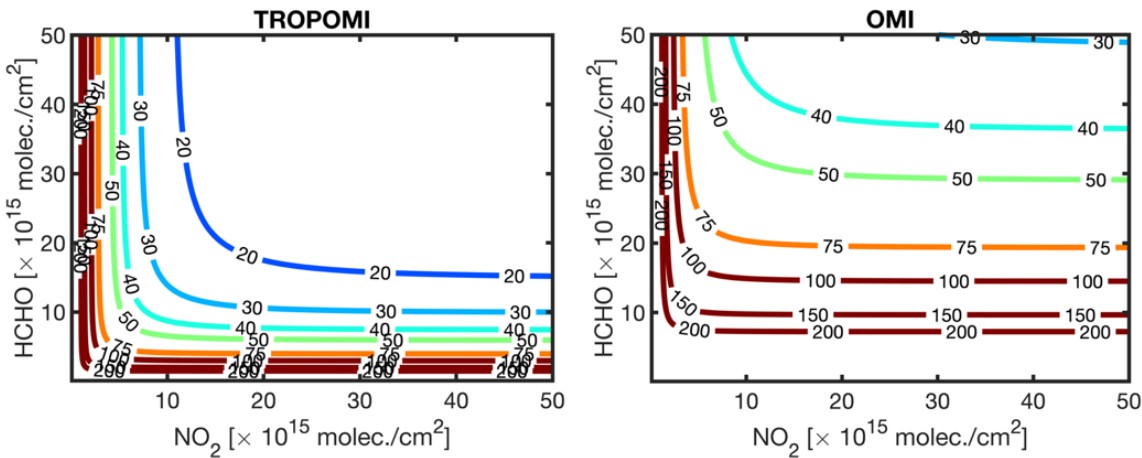

**Figure 12.** The contour plots of the relative errors in TROPOMI (left) and OMI (right) based on dispersions derived from Figure 10 and 11. The errors used for these estimates are based on daily observations.

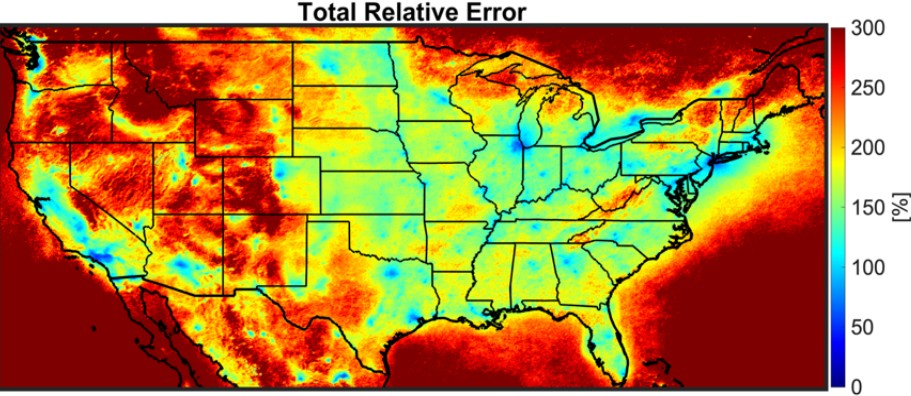

**Figure 13.** The total relative error for observed TROPOMI HCHO/NO$_2$ ratios considering the daily TROPOMI retrieval errors ($\sigma_{NO_2}$ = 2.11×10$^{15}$ molec./cm$^2$ and $\sigma_{HCHO}$ = 2.97×10$^{15}$ molec./cm$^2$), the spatial representation pertaining to OMI footprint over a city environment (13% loss in the spatial variance), and the column to the PBL translation parameterization (26%) proposed in this study. Please note that the observed FNR is based on mean values from June till August 2021, while the uncertainties used for error calculation are on daily-basis.



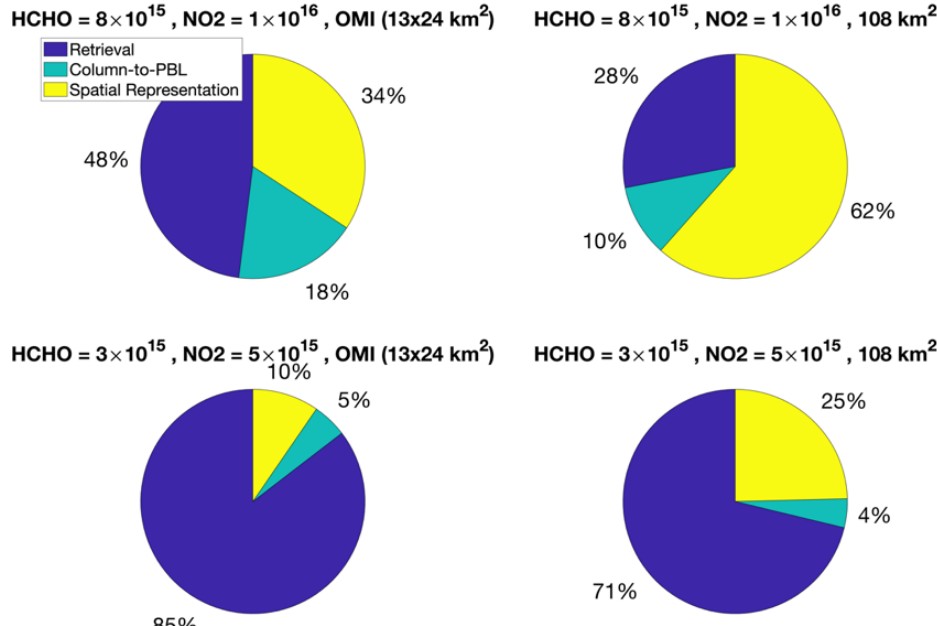

1268
**Figure 14.** The fractional errors of retrieval (blue), column to PBL translation (green), and spatial
representation (yellow) of the total error budget for different concentrations and footprints based
on TROPOMI sigma values. The retrieval error used for the error budget is on daily basis.



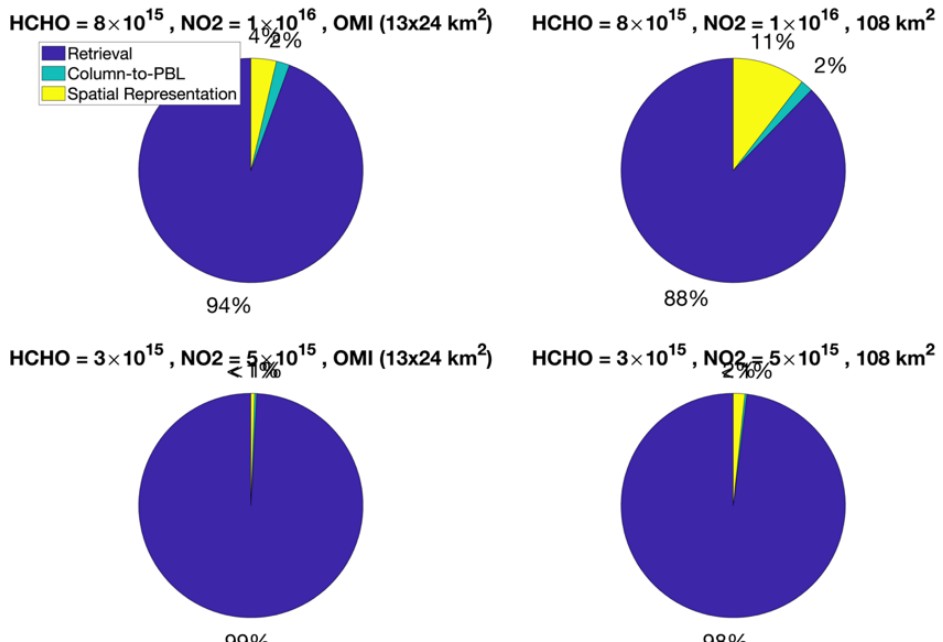

1272
**Figure 15.** Same as Figure 14 but based on OMI sigma values.
1274