# Peer review of "Characterization of Errors in Satellite-based HCHO/NO2 1 Tropospheric Column Ratios with Respect to Chemistry, Column to 2 Translation, Spatial Representation, PBL and Retrieval 3 Uncertainties 4"

_Atmospheric Chemistry and Physics, 2022_

## Author Response (AR1)

This manuscript presents a detailed and comprehensive analysis on the use of the HCHO/NO2 as measured by satellites to characterise the photochemical regimes for ozone production. The manuscript focusses on four different aspects usefulness of HCHO/NO2 as a proxy, the impact of the vertical distribution, spatial heterogeneity, and retrieval uncertainties itself. The analysis draws from a range of model and measured data and makes uses of different statistical approaches. The manuscript provides a wealth of information, but it will be most valuable for the specialist community. I recommend publication in Atmos. Chem. Phys. (although it would also fit well into AMT) after consideration of my comments below.

For the different aspects, different methods and different statistical metrics are used. I would like to get some justification why a specific metric is used and more detail on applied the methods:

| Answer |
| --- |
| **We thank the reviewer for taking their time to provide constructive comments. Our response follows:** |

- Altitude dependency (section 3.5)

  o Can you please provide some more details on the equation used to compute the first moment of the area (equation 9). The moment of an area is the integral of distance over area. Also, dz is missing.

| Answer |
| --- |
| **Thanks for noticing this. We added "dz".**

 **We also elaborated about the notion of the formula.**

 **If we rotate the vertical distribution in HCHO/NO$_2$ in Fig. 5 by 90-degree in counter-clockwise direction, we will have the new x-axis as height (z), and the y-axis as the ratio. The centroid along the y direction (the ratio) can be obtained through $1/A \int y\ (x2-x1)\ dy$ or alternatively $1/2A \int (y2)\wedge 2 - (y1)\wedge 2\ dx$. Reversing the dimension ($x \rightarrow z$ and $y \rightarrow$ HCHO/NO$_2 \rightarrow f(z)$), we get Eq.9.** |

| Modifications |
| --- |
| We added to the text that: "One can effortlessly fit this function to different bounds of the vertical distribution of FNR such as the $25^{th}$ and $75^{th}$ percentiles, and subsequently estimate the first moment of the resultant polygon along $z$ divided by the total area bounded to the polygon via:"

 $$G(z_1, z_2) = \frac{1}{2A} \int_{z1}^{z2} f^2(z)_{75th} - f^2(z)_{25th}\ \mathrm{dz}$$ |

  o Note that a satellite observesa column which is either given by the integral of the concentration over altitude or mixing ratio over pressure, while here mixing ratios seem to be integrated over height which is not correct.

**In satellite observations, we prefer to use pressure (or air density along with mixing ratios) because, in the spectral fitting, we usually estimate the total number of molecules. As a result, it is more convenient to present the concentrations as partial columns or to present the vertical mixing ratios as function of air pressure. In the integral, we calculated the mass centroid of the ratio of HCHO/$NO_2$ bounded to different percentiles. The ratio is unitless, independent of the unit choice (partial columns or mixing ratios). Therefore, readjusting the equations using the air pressure instead of height should not change any result. However, we wish to keep the formula as a function of height because the *adjustment factor* is suitable when applied for a *given PBL height* which is conventionally given as *km*.**

o   Why is the standard-deviation of the ratio of the first moment of the interquartile range a good metric for the uncertainty

**There are three uncertainties associated with the adjustment factor: i) the fact that we use observations from only a few campaigns limits our ability to say that the adjustment factor can be generalized to everywhere and every time. We discuss this caveat in detail in the original version of the manuscript. Also, ii) the boundary choice for Eq.9 is subjective. We had tested it for various numbers and as we mentioned in the paper, the adjustment factor became unrobust for large percentiles (> 80[th] percentile). Finally, iii) there is an error in our assumption about using second order rational functions to describe the vertical distributions in the ratio. This is where the 26% error comes from. We re-estimated the adjustment factors for different coefficients at 1 sigma level (68% confidence level) in Eq.8 (the second-order rational functions) to be able to create the dashed red line in Figure 6. We have elaborated this in the new draft.**

**We added that:**

"…where *zt* can be interchanged to match the PBLH. This definition is more beneficial than using the entire tropospheric column to the surface conversion (e.g., Jin et al., 2017) because ozone can be formed in various vertical layers. To determine the adjustment factor error, we reestimate Eq.9 with ±1σ level in the coefficients obtained from Eq.8. The resultant error is shown in the dashed red line in Figure 6. This error results from uncertainties associated with assuming that the second-order rational function can explain the vertical distribution of FNRs."

o   What is the impact of altitude sensitivity of the satellite column measurement as described by the averaging kernel on the estimate uncertainty.

**This is a great comment. We agree that the magnitude of sensitivity of the radiance to optical thickness within the wavelengths used for HCHO (~350 nm) and $NO_2$ (~450 nm) is not the same. HCHO tends to have a lower sensitivity to the tropospheric region making VCDs more dependable on the prior model information (AMF). But one of the biggest motivations of using the ratio as described in Martin et al., 2004, and our very recent study in AMT (Johnson et al., 2022), is the fact that the shape of scattering weights is not too drastically different for these two channels. As a result, the first-order discrepancy in scattering weight calculation get normalized after we divide HCHO VCDs by NO2 VCDs. This is why the mean bias in the ratio gets closer to**

zero in (Johnson et al., 2022: https://amt.copernicus.org/preprints/amt-2022-237/ ), despite the fact that individual products can possess a large mean bias.

Regarding the adjustment factor, the **shape** of scattering weights only matter (rather than the absolute values) which is not drastically different for those two bands (for a generic land pixel) within the first 5 km where the largest variability in the ratio lies in. See Figure 4 in https://amt.copernicus.org/articles/10/759/2017/amt-10-759-2017.pdf. Or the blue line in Figure 2 in https://amt.copernicus.org/articles/11/5941/2018/ within 5 km (~ 600 mbar). So we do not think it will introduce a larger inhomogeneity in the columnar ratio. Our assumption may not hold for a particular scene with variable extinction efficiency induced by complex aerosol optical properties between 350 and 450 nm, or for a specific viewing geometry (particularly, when the geometric AMF is large around early morning or late afternoon). So we added a caveat saying that we had assumed that the shape (the curvature) of the scattering weights of HCHO and $NO_2$ between surface and 5 km (around 600 mbar) is rather similar.

| Modifications |
|---|

We added the caveat:

"A lingering concern over the application of satellite-based FNR tropospheric columns is that the vertical distribution of HCHO and $NO_2$ are integrated in columns thus this vertical information is permanently lost. As such, here we provide insights on the vertical distribution of FNR within the tropospheric column. This task requires information about the differences between i) the vertical shape of HCHO and that of $NO_2$ and ii) the vertical shape in the sensitivity of the retrievals to the different altitude layers (described as scattering weights). Ideally, if both compounds and the scattering weights show an identically relative shape, the FNR columns will be valid for every air parcel along the vertical path (i.e., a straight line). Previous studies such as Jin et al. (2017) and Schroeder et al. (2017) observed a large degree of vertical inhomogeneity in both HCHO and $NO_2$ concentrations suggesting that this ideal condition cannot be met. A real-time true state of their vertical distribution is not always present, but a natural way of accounting for their distribution is to use retrospective measurements to constitute some degree of generalizations. As for the differences in the vertical shapes (i.e., the curvature) of the sensitivity of the retrievals between HCHO and $NO_2$ channels (i.e., ~ 340 nm and ~440 nm), under normal atmospheric and viewing geometry conditions, several studies such as Nowlan et al. 2018 and Lorente et al. 2017 showed small differences in the vertical shapes of the scattering weights within first few kilometers altitude above the surface where the significant fluctuations in FNRs usually take place. Therefore, we do not consider the varying vertical shapes in the scattering weights in our analysis. This assumption might not hold for excessive aerosol loading with variable extinction efficiency between ~340 nm and ~440 nm wavelengths or extreme solar zenith angles."

- Spatial heterogeneity (Section 3.6)

  o Please justify the use of the metrics given in equation 14 to quantify the representation error.

  o Important to point out that this is not an absolute but a relative metric (with 3x3 km2) as reference

| Answer |
|---|

The spatial information or variance can be described by the spatial autocorrelation or semivariogram described in Eq. 12 (Matheron, 1963). Our previous study showed how this operator can describe the level of spatial heterogeneity or variance in idealized cases in Figure 1 in https://amt.copernicus.org/articles/15/41/2022/.  The semivariogram can be influenced by noise. As a result, we need to fit a function to the semivariogram such as the stable Gaussian distribution used in Souri et al., 2022. The modeled semivariogram then can be used to compare

one dataset to another one allowing for understanding the extent of the spatial variance at a specific length scale each field provides. If two fields show an identical spatial variance (say the first field has a plume, and the second field has the identical plume but rotated 90 degree clockwise), both semivariograms will be identical and the ratio of $\gamma$ to $\gamma_{ref}$ will be 1 meaning our target can 100% represent the spatial variance presented in the reference. The ratio of $\gamma$ to $\gamma_{ref}$ cannot go above 1 as long as we base the reference on a finer dataset (the baseline). So Eq 14 (1 – gamma/gamma_ref) is proposed to calculate the opposite effect meaning how much of information the target field has lost compared to the reference.

We also added the caveat saying the metric is metric.

| Modifications |
| --- |
| **We added:**

"To remove potential outliers (such as noise), it is wise to model the semivariogram using an empirical regression model. To model the semivariogram, we follow the stable Gaussian function used in Souri et al. (2022).."

"where $\gamma(h)$ and $\gamma_{ref}(h)$ are the modeled semivariogram of the target and the reference fields (3×3 km²). This equation articulates the amount of information lost in the target field for the reference. Accordingly, the proposed formulation of the spatial representation error is relative." |

- Satellite errors (section 3.7):
  - 15 assumes uncorrelated random errors between the HCHO and N02 retrieval. This is the case of measurement noise-driven errors but the scatter (standard deviation) in both will also be the result of variable geophysical parameters (e.g. aerosols) which will have some level of correlation.

| Answer |
| --- |
| **We agree. We added a caveat.** |

| Modifications |
| --- |
| **We added:** "where $\sigma_{HCHO}$ and $\sigma_{NO_2}$ are total uncertainties of HCHO and $NO_2$ observations. It is important to recognize that the errors in HCHO and $NO_2$ are not strictly uncorrelated due to assumptions made in their air mass factor calculations. The consequence of disregarding the correlated errors is an underestimation in the final error. " |

  - What is the role the different averaging kernels between the satellite and ground-based DOAS instruments

| Answer |
| --- |
| **That's an excellent comment which has been mentioned in Verhoelst et al., 2021.** |

| Modifications |
| --- |
| **We added:** "Verhoelst et al. (2021) rigorously studied the potential root cause of some discrepancies between MAX-DOAS and TROPOMI. An important source of error stems from the fundamental differences in the vertical sensitivities of MAX-DOAS (more sensitive to the lower tropospheric region) and TROPOMI (more sensitive to the upper tropospheric area)." |

- Total error (Section 3.8)
  - The different error terms are combined into a total error. However, only assumed random components of uncertainties are included (and not systematic ones) so it should be called the total random error. For me, eq. 16 is too some extend trying to combine apples and oranges as the underlying metric in the 3 components is very different and have different meanings.

**Answer**

Concerning the random error versus systematic errors, we disagree that the sigma values obtained from the histograms are purely random. As a matter of fact, the biggest portion of these errors originate from unresolved systematic errors (or relative errors) in the retrievals. For instance, a relative bias error in the surface albedo manifests in varying biases (systematic) in the retrieval resulting in a large dispersion in the histograms. To better demonstrate this, we recreated the TROPOMI-MAXDOAS histogram based on the monthly-basis observations (Figure S9), and we observed the standard deviation barely changing. If the dispersion were purely random, we would see them going down by 1/sqrt(number_samples). This is why in the beginning of this section, we stated the errors were total uncertainties. Because we do not repeat the same experiment, and the underlying root cause of the errors is unknown, it is impossible to single out the systematic errors from the random errors. The median in the histogram only explains the median of the systematic biases which can vary from pixel to pixel.

[Figure]

In terms of combining different errors, we agree that each error is different in nature, but they all correspond to one quantity, the ratio. Each error explains the extent of information in the satellite-based columns that can become unavailable. Therefore, the sum of them is a good metric to know the combined error. In the satellite retrieval, it is common to have an additive squares errors of different errors such as the temporal representativity, the spatial representativity, the smoothing error, the aggregation error, and the radiative transfer model (RTM) error parameter (Rodgers, 2000). Each of these individual error components come with different meaning but they are translated to one unified quantity. To be able to compare the magnitude of these errors to each other, it is critical to know the combined error which we assumed that their squares are additive. This helped us recognizing the fact that the retrieval error is the largest obstacle in applying the ratio in a robust manner. A very recent group from the same project observed large errors in the satellite retrievals (**https://amt.copernicus.org/preprints/amt-2022-237/**).

**Modifications**

To elaborate on the random vs systematic errors we added:

"This fitted normal distribution ($R^2$=0.94) is used to approximate $\boldsymbol{\sigma_{NO_2}}$ for different confidence intervals and to play down blunders. To understand how much of these disagreements are caused by systematic errors as opposed to random errors, we redo the histogram using monthly-based observations (Figure S14). A slight change in the dispersions between the daily and the monthly-basis analysis indicates the significance of unresolved systematic (or relative) biases. This tendency suggests, when conducting the analysis on a monthly basis, the relative bias cannot be mitigated by averaging. Verhoelst et al. (2021) rigorously studied the potential root cause of some discrepancies between MAX-DOAS and TROPOMI. A important source of error stems from the fundamental differences in the vertical sensitivities of MAX-DOAS (more sensitive to the lower tropospheric region) and TROPOMI (more sensitive to the upper tropospheric area). This systematic error can only be mitigated using reliably high-resolution vertical shape factors instead of spatiotemporal averaging of the satellite data."

**Regarding the total error,**
       "The ultimate task is to compile the aforementioned errors to gauge how each individual source of error contributes to the overall error. Although each source of error is different in nature, combined they explain the uncertainties of one quantity (FNR) and can be roughly considered independent; therefore, the combined error is given by:"
       **We also changed the total error to the combined error to emphasize that this is simply a linear combination of error:**
       "To build intuition in the significance of the errors above, we finally calculated the combined error in the ratio by linearly combining the root sum of the squares of the TROPOMI retrieval errors, the…"

**We also mentioned our new study too:**
"This experiment suggests a standard deviation of $9.4 \times 10^{15}$ molec./cm$^2$ with which we again observe the retrieval error to be the largest contributor (>80%) of the combined error (Figure S10). A recent study (Johnson et al., 2022) also suggests that retrieval errors can result in considerable disagreement between FNRs between various sensors and retrieval frameworks."

Minor points:

- Please make sure that all acronyms and abbreviations are spelled out when used for the first time (e.g. NOx, P(O3), DISCOVER-AQ, PAN, VOC, SENEX, SZA, …)

| Answer |
| --- |
| **Sure. We reread the draft and made sure they are spelled out.** |

- 4, l149: …FNR from **a** chemistry perspective…

| Answer |
| --- |
| **Done.** |

- 5, l.188: heterogenous chemistry is not considered -> can you add a statement on the importance of that assumption on the study.

> **Answer**
>
> **We already mention this is not a major concern in the original version of the manuscript:**
>
> "Brune et al. (2021) provided compelling evidence showing that the consideration of the $HO_2$ uptake would make the results significantly inconsistent with the observations suggesting that the $HO_2$ uptake may have been inconsequential during the campaign. "

- 5, l.206: hv -> h. and define h and (nu)

> **Answer**
>
> **Defining them will make the sentence difficult to read so we decided to remove +hv. The photolysis rates of X is meaningful enough.**

- 6, eq. 1-3: define k and M, state what the sum is summing up

> **Answer**
>
> **Defined.**

- 6, l. 239: unconstrained observations -> independent observations

> **Answer**
>
> **Changed.**

- 6 l. 255: contrary to **an** overestimation in clean ones

> **Answer**
>
> **Changed.**

- 7, l.262: of NO in **the** chemical mechanism

> **Answer**
>
> **Changed.**

- 7, l.262: some of **the** oxygenated VOCs

> **Answer**
>
> **Changed.**

- 7 l264: with larger PAN because -> with larger PAN **mixing ratios** because

| Answer |
|---|
| **Changed.** |

- 7, l.277: to reproduce HO2 with -> to reproduce HO2 **mixing ratios** with

| Answer |
|---|
| **Changed.** |

- 7, l. 286: 0.62 $10^6$ cm-3 -> 0.62 x $10^6$ cm-3

| Answer |
|---|
| **Changed.** |

- 7 l. 288: at least virtually representative -> what do you mean by 'virtually'?

| Answer |
|---|
| **We meant roughly. Changed to roughly.** |

- 7, l. 291: an analytical solution suggesting… -> solution to what?

| Answer |
|---|
| **We removed it.** |

- 8, l. 328: PO3 -> this has been written as P(O3) before.

| Answer |
|---|
| **We standardized them as $PO_3$.** |

- 10, l.399-402: I don't clearly see this larger decrease in NO2 than of HCHO. The media value of the ratio in Fig.5 is more or less 5 with some variability.

| Answer |
|---|
| **Up to 5 km, the median moves to higher values. This tendency has been well documented in Jin et al., 2017 and Schroeder et al., 2017.** |

**If we combine the data into 1 km layer thickness, the trend will stand out clearly but we want to also keep the variability:**

[Figure]

- 31: figure :3 the 3 green lines are very hard to distinguish.

| Answer |
| --- |
| **Thanks we have changed the color.** |

- 37, Figure 9: I assume the y-axis is not given in %

| Answer |
| --- |
| **Yes, we forgot to multiply to 100. Corrected.** |

Souri et al present a detailed study highlighting four major shortcomings associated with FNRs and their ability to categorize ozone sensitivity. The sections about column-to-PBL translation, spatial representation error, and retrieval error are all well-written. The manuscript as a whole has understandable writing style and clear, well-made figures. However, I do have a few major concerns, mostly surrounding the modeling section of this work. I recommend that the manuscript be sent to the authors for major revisions.

| Answer |
| --- |
| **We thank the reviewer for taking their time to provide constructive comments. Our response follows:** |

**VOC inputs for the box model:** The modeled radical environment can be incredibly sensitive to changes in VOC inputs, especially in polluted urban areas. This manuscript is lacking detail about how VOC inputs were created, leaving readers to assume the authors used a simplistic approach that excludes many potentially important VOCs. As written, the authors' treatment of VOC inputs does not rise to the level established in previous modeling studies performed for the same field campaigns, leaving this reviewer wondering if the modeling presented in this study can represent the ambient radical environment.

The field campaigns modeled in this study have unique VOC measurement suites which require unique data engineering strategies to generate realistic VOC inputs. DISCOVER-AQ was served only by a quadrupole PTRMS, and features a very limited set of VOCs. The authors do not give adequate detail about how they generated VOC inputs based on these data. For example, previous studies (i.e. Schroeder et al 2017) generated speciated VOC box model inputs for DISCOVER-AQ using a fusion of VOC data from concurrent airborne campaigns (DISCOVER-AQ+SEAC4RS+FRAPPE). This enabled somewhat realistic estimation of VOCs that were not measured by the PTRMS during DISCOVER-AQ.

During KORUS-AQ, the whole air sampler was flown concurrently with a PTRMS, giving a richer suite of speciated VOCs. However, these two instruments had wildly different sampling cadences and integration times, with WAS measurements being incapable of resolving fine-structure details in pollutant gradients. As a result, previous studies (i.e. Schroeder et al 2020) fused the two datasets together to generate a pseudo-high-resolution set of VOC inputs for their box modeling work with KORUS-AQ.

As it is currently written, I have serious concerns about the VOCs used as model inputs, and thus have lower confidence in the results presented here. Can you show that the simplistic VOC inputs used in this study do not yield significantly different results from the two Schroeder papers?

Perhaps a more pointed observation: the box model inputs and outputs from the two Schroeder papers are publicly available online. What does this study gain by running its own model simulation – with questionable VOC representation – instead of using the freely-available Schroeder/Crawford data which has already been heavily vetted and used in multiple studies?

| Answer |
| --- |
| **We fully understand the reviewer's concern regarding the gaps associated with the VOC measurements and that each field campaign provides a different set of measurements. When setting up the model, we factored in two criteria about how we should go about the VOC treatment (which includes both measurements and the model chemical mechanism). Two major criteria:**
    1.  **How much difference does a specific chemical mechanism make in terms of OH, HO$_2$, and HCHO, given a fixed number of measured VOCs?** |

We have tested the GEOS-Chem v12 chemical mechanism with an update to aromatic VOCs based on Bates et al., 2021, MCM, and the well-established CB06 mechanism. Based on our observations, we realized that the CB06 mechanism could simulate results similar to the MCM but at a much cheaper computational cost. Unlike previous studies, we feel it is better to not have HCHO constrained so that we can truly understand how much variance (information) in the observations each model realization can replicate, given the chemical mechanism and measurements used. We concluded that the selected VOCs (Table 1) are sufficient to replicate observations with more than 70% variance in HCHO. We also noticed that the performance of radicals such as $HO_2$ and OH are highly similar compared to previous studies such as Souri et al. (2020), Schroeder et al. (2017, 2021), and Brune et al. (2021) (who also compared two different model realizations including the LaRC model used in Schroeder et al. (2021)). If the VOC treatment had been unsatisfactory, we might have observed an inferior performance in terms of HOx compared to Souri et al. (2020), Schroeder et al. (2017, 2021), and Brune et al. (2021), which is not the case (statistics had been provided in the text).

In our previous study, we compared a very similar setup with the NASA LaRC box model over a highly complex environment (Seoul, Korea) and observed a strong agreement between our model and NASA LaRC. Please see https://www.sciencedirect.com/science/article/pii/S1352231020305276

We realized that a HCHO-unconstrained version of the model output exists to test against our setup (https://www-air.larc.nasa.gov/cgi-bin/ArcView/korusaq?MODEL=1#CRAWFORD.JAMES/ ). We were hoping to conduct the same analysis with DISCOVER-Colorado AQ but we were unable to find the LaRC unconstrained simulation results on the DISCOVER-AQ archive. We synched the timetag between _input and _unconstrained files and averaged 1 Hz data to 10 sec. We conducted several sensitivity tests, including running the model with a fixed dilution factor (the original setup), without considering a dilution factor (=0), unconstraining H2O2 and HNO3, and compared the simulated HCHO with the observed ones:

[Figure]

**Figure S2. The comparison of simulated HCHO mixing ratios compared to observations for (top left) our F0AM setup with the dilution process on, (top right) same model but without the dilution process, (bottom left) our F0AM setup with dilution process on and without constraining HNO3 and H2O2, and (bottom right) NASA LaRC unconstrained model based on Schroeder et al., 2021. All points are based on 10-sec sampling size.**

**Several tendencies can be observed:**

- **All F0AM configurations capture 81% of the variance in observed HCHO with relatively low bias**
- **The dilution factor used in our study does not change the correlation coefficient or bias between simulated and observed HCHO.**
- **NASA LaRC box model captures the variance in HCHO slightly (6%) better than F0AM/CB06 but with a significant bias.**

**A valid criticism is how we can get HCHO right while leaving PAN unconstrained. When we compare the simulated PAN mixing ratios with the observations w/ and w/o dilution factor (see figure below), the results significantly changed, which could adversely impact the performance of HCHO. But when we super-imposed the aircraft altitude, we realized that those highly overestimated PAN mixing ratios mostly occurred in very high altitudes where PAN is thermally stable. It indicates that our model is skillful at reproducing PAN in lower altitudes, even without considering a dilution factor. To sum up, without considering the PAN constraint and a dilution factor, our model can still capture the magnitude and variability of HCHO which suggests that the box model results in this study are an accurate representation of ambient chemical conditions.**

[Figure]

**Figure S3. (left) The comparison of PAN mixing ratios w/ a fixed dilution factor and (right) w/o a dilution factor during KORUS-AQ campaign.**

[Figure]

**Figure S4. Same as the above right figure but with aircraft altitude superimposed.**

2) **What are the overlaps between the CB06 mechanism and the measured VOCs?**
   **The CB06 mechanism does not include several measured VOCs such as butane, hexane, and styrene. The reviewer can see the list of compounds this mechanism covers Table 5.1 in https://github.com/AirChem/F0AM/blob/master/Chem/CB6r2/CAMxUsersGuide_v6-30.pdf). We did our best to find the overlaps listed in Table1.**

**Modifications**

**We first added this part in section 2: "**Regarding the KORUS-AQ campaign where HOx observations were available, we only ran the model for data points with HOx measurements. Similar to Souri et al. (2020), we filled gaps in VOC observations with a bilinear interpolation method with no extrapolation allowed. In complex polluted atmospheric conditions such as that over Seoul, South Korea, Souri et al. (2020) observed that this simplistic treatment yielded comparable results with respect to the NASA LaRC model (Schroeder et al. 2020) which incorporated a more comprehensive data harmonization."

**We removed these two parts from section 3.1 and the conclusion, because the dilution factor did not allow the total bias to go above 5%:**
Concerning HCHO, our model does have considerable skill at reproducing the variability of observed HCHO ($R^2$=0.73) .

Our box model showed a reasonable performance at recreating some of unconstrained key compounds such as OH ($R^2$=0.64, bias=17%), $HO_2$ ($R^2$=0.66, bias<1%), and HCHO ($R^2$=0.73, ).

**We added the comparison of HCHO w/ and w/o dilution factor and with NASA LaRC:**

"Concerning HCHO, our model does have considerable skill at reproducing the variability of observed HCHO ($R^2$=0.73). To evaluate if this agreement is accidentally caused by the choice of the dilution factor and to identify if our VOC treatment is inferior compared to the one adopted in the NASA LaRC (Schroeder et al., 2021), we conducted three sets of sensitivity tests for the KORUS-AQ campaign, including ones with and without considering a dilution factor and another without $HNO_3$ and $H_2O_2$ constraints. When not considering a dilution factor results in no difference in the variance in HCHO captured by our model ($R^2$=0.81). Our model without the dilution factor is still skillful at replicating the magnitude of HCHO with less than 12% bias. It is because of this reason that the optimal dilution factor for each camping is within 12 hr to 24 hr which is not different than other box modeling studies (e.g., Brune et al., 2022; Miller and Brune, 2022). We observed no difference in the simulated HCHO when $HNO_3$ and $H_2O_2$ values were not constrained. The unconstrained NASA LaRC setup oversampled at 10-sec frequency captures 86% variance in the measurements, only slightly (6%) outperforming our result. However, the unconstrained NASA LaRC setup greatly underestimates the magnitude of HCHO compared to our model results."

**We also added the PAN comparison w/ and w/o dilution factor for the KORUS-AQ campaign:**
"Moreover, we should not rule out the impact of the first-order dilution factor which was only empirically set in this study. For instance, if we ignore the dilution process for the KORUS-AQ campaign, the bias of the model in terms of PAN will increase by 33% resulting in a poor performance ($R^2$=0.40) (Figure S3). We notice that this poor performance primarily occurs for high altitude measurements where PAN is thermally stable (Figure S4); therefore, this does not impact the majority of rapid atmospheric chemistry occurring in the lower troposphere such the formation of HCHO."

**Model Setup:** I have a few concerns with model setup:

Why use an arbitrary model run time of 5 days? Ideally, the model should be run indefinitely until it converges on a solution for key species, but I understand the desire to set a lower limit for the sake of computation. Do your outputs change if you use 4 days? Or 6 days? Or 20 days? Can you include a sensitivity analysis to back up your work – that is, show that your arbitrary choice of 5 days does not impact results?

| Answer |
| --- |
| **Thanks for the comment. Our previous study (Souri et al., 2020) used three solar cycles which resulted in almost net zero HCHO production rates. For the current study, to be conservative, we decided to increase it to five days and phrase this as "approaching" instead of "reaching" steady state. To alleviate the reviewer's concern, we ran 10 solar cycles and compared HCHO concentrations to the 5 solar cycles used in our study. The results remained identical meaning we have already reached to the steady state.** |
| Modifications |
| **We added to Sect 2: "**We test the number of solar cycles used in this study (5) against ten days on the KORUS-AQ field campaign model setup, and observe no noticeable difference in simulated OH and HCHO (Figure S1), indicating that the choice of five solar cycles suffices." |

[Figure]

Figure S1. The comparison of simulated HCHO (left) and OH (right) with 5 (y-axis) and 10 (x-axis) solar cycles.

If I understand this correctly, you calculate a unique dilution factor for each field campaign, deriving it empirically to yield the best agreement between measured and modeled HCHO. What is the physical basis for why one field campaign would have different dilution rates than another? Without further explanation, this feels like an arbitrary "correction factor" to game the model for better agreement with observations – which does nothing to tell you how well the model represents the underlying chemistry. Can you explain?

**Answer**

**The dilution factor is a highly oversimplified parameter that represents generic physical loss. It is typical, for example, to impart a 24-h lifetime to all model species to approximate physical losses due to transport, deposition, etc. Because those physical parameters vary greatly by time and space, it is logically flawed to consider a constant dilution factor for all locations/times. We chose a constant dilution factor for each campaign in order to avoid over-constraining the model by setting a dilution factor to a value for each point measurements to artificially match observations. We want to keep the box model as simple (but not overly simple) as possible to allow clear intuition about PO3 tendencies. We would like to clarify that we do not claim that using the mean bias of simulated HCHO is the best approach to understand the underlying processes of the dilution factor, as understanding many of those parameters requires the precise knowledge of land surface processes, momentum fluxes, etc. which will turn the box model framework to a full chemical transport model.**

**Schroeder et al. 2021 mentioned in their paper:** "For instance, Fried et al. (2011) demonstrated that model estimates of CH2O can be biased in pollution plumes when short-lived species dominate the model photochemistry, e.g., highly-reactive VOCs such as biogenic isoprene or alkenes from industrial sources. Under such conditions, using observations to constrain CH2O and other important radical reservoirs instead of calculating them will improve model calculations of the radical pool responsible

for ozone production." **There is a trade-off between fully constraining HCHO (to get the radicals simulation straight) while masking the understanding of if the model can actually reproduce HCHO, and keeping the mean bias of HCHO low with an optimal dilution factor while allowing HCHO to fluctuate. We chose the latter because it enabled us to have a more stringent validation on the overall model performance.**

**Given the differences in physical and meteorological conditions of the various regions of the field campaigns, one would expect that a different dilution factor for each campaign should be used. To alleviate the reviewer's concern, we performed a sensitivity test in which we entirely removed the dilution factor and reran the box model for the KORUS-AQ campaign. In this sensitivity simulation we observed negligible changes in HOx, NOx, and HCHO. PAN was obviously impacted the most; however, the vast majority of PAN overestimated values were at high altitudes where PAN is thermally stable (Figure S4).**

**Modifications**

**We have already included new figures regarding HCHO and PAN based on the first comment.**

**We added in Section 2 about the rationale behind using the dilution factor for HCHO:**

"Because the model does not consider various physical loss pathways including deposition and transport, which vary by time and space, we oversimplify their physical loss through a first-order dilution rate set to $1/86400$-$1/43200$ s$^{-1}$ (i.e., 24- or 12-hr lifetime), which in turn prevents relatively long-lived species from accumulating over time. Our decision on unconstraining HCHO, a pivotal compound impacting the simulation of HOx, may introduce some systematic biases in the simulation of radicals determining ozone chemistry (Schroeder et al., 2020). Therefore, to mitigate the potential bias in HCHO, we set the dilution factor to maintain the campaign-averaged bias in the simulated HCHO with respect to observations of less than 5%. However, it is essential to recognize that HCHO can fluctuate freely for each point measurement because the dilution constraint is set to a fixed value for an individual campaign."

**We also added the comparison of HOx w/ and w/o considering the dilution factor in the supplementary material.**

"A sensitivity test involving removing the first-order dilution process demonstrates that the simulation of HOx is rather insensitive to this parameter (Figure S5). This might be caused by the fact that the simulated HCHO already agrees relatively well with the observations without the dilution factor."

[Figure]

Figure S5. (top) The simulation of OH and HO$_2$ using a fixed dilution factor during the KORUS-AQ campaign. (bottom) without considering the dilution factor.

**Model Validation:** As written, the model validation section does not give me confidence in the model's ability to represent the ambient radical environment (especially given the simplistic treatment of VOCs).

If one of the model parameters (dilution rate) is based on empirical model/measurement agreement, then comparing simulated values to observations is cyclical. If I am understanding this correctly, then Section 3.1 is incredibly problematic. Based on the description given in line 220, campaign-average simulated HCHO is not allowed be >5% off from campaign-average observed HCHO – its part of the model setup with an empirically derived dilution factor. How can you evaluate the model's representation of the chemical environment with such a setup? For example, in line 254 you state that HCHO had a mean bias of less than 5% - which is meaningless because you've coded the model to do exactly that.

In practice, your dilution factor acts as a quasi-constraint on HCHO, which greatly influences calculated radical budgets. This eliminates your ability to truly test whether the model is capable of representing the radical budget from first principles. Furthermore, this does not allow you to test if your simplistic treatment of VOCs is adequate. You mention that the bias in PAN changes if you ignore dilution, but PAN can have a large impact on modeled radical and NO2 concentrations. What happens to other test-species if you ignore dilution?

I don't buy the idea that this model, as currently setup, has proven itself as sufficient for representing ozone chemistry.

| Answer |
| --- |
| **We have addressed this valid concern in the previous comment. Our results are insensitive to the dilution factor.** |
* * *
In short, you have questionable VOC inputs and a questionable model setup which prevents you from truly testing model performance. I'd suggest re-running the model with a fixed dilution factor based on reasonable physics. This will enable "true" unconstrained model runs, providing a testbed for evaluating model performance (and your VOC inputs). Or, you could run another, established model in parallel on a subset of data and compare the two. Or, you could use the freely-available model inputs/outputs from published studies from the same field campaigns, rather than re-invent the wheel.

| Answer |
| --- |
| **We showed that ignoring a dilution factor did not broadly impact the biases in HOx and HCHO compared to our baseline simulations in the study. We want to clarify that we did not apply a variable dilution factor for each point measurement to over-tune the model. A fixed dilution factor for each campaign did not impact the variability in simulated HOx and HCHO. The most significant impact caused by the dilution factor on our result was seen for PAN in high altitudes, whose effects on HOx and HCHO were inconsequential. Comparing the unconstrained NASA LaRC model output with our model during KORUS-AQ showed that both models were skillful at replicating HCHO ($R^2$=0.87 vs. $R^2$=0.81). Even so, our model shows less significant biases in HCHO than the NASA LaRC model.**

 **Regarding the comment about "re-inventing the wheel" by not using the LaRC box model, we have strong reasons for not doing so:**

 1- **It is in our opinion that that while the NASA LaRC publicly available (not sure if it applies to DISCOVER-AQs) outputs are great, not having the capability to run, modify, update, perform sensitivity tests, receive technical and scientific helps from the steering committee and peers (such as for publicly available models such as F0AM (0-D), WRF, CMAQ, GEOS-Chem, CESM, and MUSICA) is a disadvantage for conducting research. For instance, we would not be able to conduct simple sensitivity tests requested by a reviewer, such as we have done here, when we cannot run the actual model. Furthermore, if a reviewer asked about chemical sinks or sources of HCHO, or individual terms of PO3 that are not saved in the outputs, it would be impossible to address such requests.**
 2- **The F0AM model has been extensively used in more than 72 published studies with notable citations. This is a testament to open-access and transparent models. Plus, as this is NASA funded research it is important for the purposes of "open science" it is vital to use a publicly-available model to allow reproduction of our results.**
 3- **A major concern of ours for running a box model was the choice of chemical mechanism. The F0AM model provides the flexibility to easily test and apply different well-established chemical mechanisms or update individual reactions. We did not include those sensitivity tests in the paper, because we do not want to overwhelm readers of this study.** |
* * *
**Purpose of the paper:** Finally, I would challenge the authors to include paragraphs in the Introduction and Summary sections describing the motivation for doing this work. Why are incremental improvements in our understanding of FNRs and ozone chemistry necessary? Martin et al first published their paper about satellite FNRs more than twenty years ago – yet, to the best of my knowledge, no regulator or policymaker has ever used satellite FNRs in their ozone planning strategies. Clearly FNRs were first developed as a potential tool for policymakers to fine-tune ozone mitigation strategies, but if policymakers have shown no interest in using these tools, why continue refining them? Is there a pathway for FNRs to be used by anyone outside of academia? What does the author think is preventing policymakers from using this tool - or is this simply a tool for academics?

| Answer |
|---|
| **This study was designed specifically to tackle the limitations of using satellite-based FNRs and directly answers why the satellite-based FNRs (or, in general, satellite trace gases) have not been widely used by regulators. The purpose of this paper, as explicitly mentioned in the title, abstract, etc., is to quantify the errors associated with FNR rather than to refine the metric. To our knowledge nowhere in the manuscript do we suggest that satellite FNR values are sufficient for regulatory purposes. This study, for the first time, quantifies the main errors/uncertainties associated with satellite column trace gas retrievals and resulting FNRs. We argue these results could be used by regulatory agencies for a quantitative understanding of why satellite data is currently not sufficient for application in developing/testing ozone mitigation strategies.**

 **While we think it is essential to share the limitations and challenges associated with using satellite products with air quality regulators, we believe it will be well out of the ACP focus, which is "*on studies with important implications for our understanding of the state and behaviour of the atmosphere*". Furthermore, the funding agency for this work, NASA, is a Research & Analysis (*R&A*) organization and is not focused on regulatory processes. Providing quantitative information about satellite capabilities for air quality research is the primary driver of this research and the results are directly applicable for regulators. Some limitations of satellites identified in this study have been overlooked for years and not mentioned in any studies back to Martin et al. Our study is an eye-opener for the retrieval community as our work demonstrates that retrieval errors must be reduced for satellites to be applicable for regulations.** |

---

## Author Response (AR2)

**Comments to the author**:
Dear Amir Souri,

I'm pleased to accept your revised manuscript "Characterization of Errors in Satellite-based HCHO/NO2 Tropospheric Column Ratios with Respect to Chemistry, Column to PBL Translation, Spatial Representation, and Retrieval Uncertainties" for ACP subject to minor revisions as explained below.

The detailed replies to the comments made during the review are appreciated, and this discussion will be a valuable resource for the interested readers.

| Answer |
| --- |
| **We thank the editor for carefully reading our manuscript/replies and helping us bring our draft to a higher standard.** |

One aspect of the results remains confusing to me: In Figure 6, you show a correction factor between the columnar HCHO/NO2 ratio and the corresponding ratio for a column from the surface up to a certain height. For the lowest kilometre, this correction factor has a value of 0.6 However, when looking at Figure 5, I do not see any evidence in the median values for the need of such a factor. With the exception of the values in the highest layers (which have only a very small contribution to the columnar ratio observed from satellite, the ratio appears very stable between 4 and 6. Please explain why the correction factor is so large close to the surface.

| Answer |
| --- |
| **We are grateful for this comment because we realized that there was indeed a missing statement in our code to compute the adjustment factor. Although the violin plots represent the afternoon observations, we forgot to apply such conditions when calculating the centroids. As a result, the adjustment factor wrongly considered both morning and afternoon times. The gradients in the ratio tend to be steeper in the morning times (from 2 close to the surface to 8 in 5-6 km) resulting in smaller adjustment factors for areas close to the surface (See Jin et al., 2017). The new adjustment factor fluctuates closer to 1.0. Please note that we fit the second-order rational functions to the 25th and 75th percentiles to create the bounding polygon used for the centroid calculation.** |

[Figure]

**Because the contrast between morning and afternoon results is interesting, we decided to add new figures to the SI regarding this tendency.**

Modifications

**We replaced Figure 6 and Figure S12. We now see 19% standard deviation around the optimal fit as opposed to 27% error in the previous results (computed as the average of the deviation of the 1-sigma from the optimal fit throughout the altitudes). Therefore, we recreated Figure 13, 14, 15, and Figure S16. These changes did not impact the conclusion of our study.**

**We modified the error:**

"The standard error deviation of this conversion is around 19%."

"This data-driven adjustment factor exclusively derived from afternoon aircraft profiles during warm seasons in non-convective conditions had a standard error of 19%."

**We recalibrated the empirical equation to represent the adjustment factor:**

It is beneficial to model this curve to make this data-driven conversion easier for future applications. A second-order polynomial can well describe ($R^2$=0.97) this curve:
$$f_{\text{adj}} = az_t^2 + bz_t + c \qquad\qquad a = -0.01, b = 0.15, c = 0.78 \qquad\qquad (11)$$

**We added in the results and discussion:**

"The relatively low fluctuations in the adjustment factor around one suggest that under the observed atmospheric conditions (clear-sky afternoon summers), the columnar tropospheric ratios do not poorly represent the chemical conditions in the PBL region."

**We added new figures to the supplemental material:**

A caveat with these results is that our analysis is limited to afternoon observations because we focus on afternoon low-orbiting sensors such as OMI and TROPOMI. Nonetheless, Schroeder et al. (2017) and Crawford et al. (2021) observed large diurnal variability in these profiles due to diurnal variability in sinks and sources of $NO_2$ and HCHO and atmospheric dynamics. The diurnal cycle has an important implication for geostationary satellites such as Tropospheric Emissions indeed: Monitoring of Pollution (TEMPO) (Chance et al., 2019). Limiting the observations to morning time results in a smaller adjustment factor for altitudes close to the surface resulting from steeper vertical gradients of $HCHO/NO_2$ (Figures S13 and S14). This tendency agrees with Jin et al. (2017), who observed a large deviation from one in an adjustment factor used for the column-surface conversion in winter.

[Figure]

Figure S13. The adjustment factor defined as the ratio of the centroid of the polygon bounding 25[th] and 75[th] percentiles of the observed $HCHO/NO_2$ columns by the NASA aircraft between the surface to 8 km to the ones between the surface and a desired altitude. This factor can be easily applied to the observed $HCHO/NO_2$ columns to translate the value to a desired altitude stretching down to the surface (i.e., PBLH). Only observations made during morning are used.

[Figure]

Figure S14. The violin plots of the morning vertical distribution of HCHO, NO2, and HCHO/NO2 observations were collected during DISCOVER-AQ Texas, Colorado, Maryland, and KORUS-AQ campaigns. The violin plots demonstrate the data distribution (i.e., a wider width means a higher frequency). White dots show the median. A solid black line shows both the 25th and 75th percentiles. The heatmap denotes the simulated ozone production rates.

**In conclusion we added:**

This behavior means that the ozone regime tends to get pushed slightly towards the VOC-sensitive regime near the surface for a given tropospheric columnar ratio. This tendency was more pronounced in morning times when the non-linear shape of FNRs was stronger.

In the discussion of the larger scatter of the HCHO values, I do not fully agree with your formulation on the absorption strengths of HCHO and NO2. The differential absorption which is used in DOAS is actually larger for HCHO than for NO2 in the UV. The NO2 differential absorption cross-section is much larger at visible wavelengths which is why NO2 retrievals usually are performed at wavelengths > 400 nm. An additional important factor is the smaller radiance and thus larger photon shot noise in the UV.

| Answer |
| --- |
| **We agree that for very small short wavelengths, the HCHO optical depth can become stronger than NO2; but we meant in the wider range of UV-ViS range, the optical depth of NO2 is much stronger. We agree that SNRs are larger for higher wavelengths providing higher fidelity information for $NO_2$ retrievals compared to HCHO.** |
| **Modifications** |

> **We rewrote the sentence to: "**The distribution is slightly broader compared to that of $NO_2$, manifested in a larger standard deviation $4.32\times10^{15}$ molec./cm$^2$. This is primarily due to two facts: i) HCHO optical depths generally peak in the UV range (<380 nm), where the large optical depths of ozone and Rayleigh scattering result in weaker and noisier signals (Gonzalez Abad et al., 2019), and ii) the broader and stronger $NO_2$ optical depths in the ViS range (400-500 nm), where the signal-to-noise ratio is typically more outstanding, permit better quality retrievals."

In Figure 7, lowest panel I'm wondering what exactly is shown here - is that the mean of the 90 individual ratios, or is it the ratio of the mean values? Please clarify in the figure caption.

| Answer |
| --- |
| **It is the ratio of the mean values shown in other panels. We do not necessary have 90 ratios as clouds can substantially reduce the number of samples within the time frame.** |
| **Modifications** |
| **We added:**

"Figure 7. Oversampled TROPOMI total HCHO columns (top), tropospheric $NO_2$ columns (middle), and the ratio (bottom) at $3\times3$ km$^2$ from June till August 2021 over the US. The ratio map is derived from the averaged maps shown in the top and middle panels." |

There still are many sentences in the manuscript which to me as a non-native speaker appear to have language issues. I have marked some of them in the attached file; please check again.

| Answer |
| --- |
| **We gave the draft a very good read and adjusted poor grammar.** |

Best regards,

Andreas Richter